
# Two infinite families of facets of the holographic entropy cone

Bartlomiej Czech[⋆], Yu Liu[†] and Bo Yu[‡]

Institute for Advanced Study, Tsinghua University, Beijing 100084, China

⋆ bartlomiej.czech@gmail.com , † yuliu21@mails.tsinghua.edu.cn ,
‡ yu-b20@mails.tsinghua.edu.cn

## Abstract

We verify that the recently proven infinite families of holographic entropy inequalities are maximally tight, i.e. they are facets of the holographic entropy cone. The proof is technical but it offers some heuristic insight. On star graphs, both families of inequalities quantify how concentrated/spread information is with respect to a dihedral symmetry acting on subsystems. In addition, toric inequalities viewed in the K-basis show an interesting interplay between four-party and six-party perfect tensors.

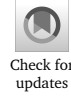
# 1   Introduction

Recent progress in the AdS/CFT correspondence [1] strongly suggests that the quantum origin of spacetime geometry involves quantum information theory. A cornerstone of this relation is the Ryu-Takayanagi (RT) formula [2, 3] and its generalizations [4–7], which relate extremal areas in spacetime to entanglement entropies of the corresponding quantum state in the holographically dual theory.

However, certain non-generic quantum states—for example, the GHZ state on four or more parties—have entanglement entropies, which cannot be consistently interpreted as Ryu-Takayanagi surfaces [8, 9]. To gain insight into what makes entropies into areas, one would like to characterize quantum states whose subsystem entropies are **marginally** consistent with an RT-like interpretation. Entropies of such states are said to live on **facets** of the **holographic entropy cone**. This paper establishes, for the first time, two infinite families of facets of the holographic entropy cone.

**Nomenclature**   Linear conditions under which subsystem entropies can be promoted to minimal areas are collectively said to define the **holographic entropy cone**. Indexing such conditions with $i$, we can exhibit the cone in the form:

$$\forall i: \quad \text{LHS}_i \geq \text{RHS}_i \quad \Rightarrow \qquad \text{consistent with RT.} \tag{1}$$

By contrapositive, we also have:

$$\exists i: \quad \text{LHS}_i < \text{RHS}_i \quad \Rightarrow \quad \text{not consistent with RT.} \tag{2}$$

In (1) and (2), quantities $\text{LHS}_i$ and $\text{RHS}_i$ are linear combinations of subsystem entropies $S_X$ with positive coefficients. The origin of the nomenclature is that a collection of linear conditions such as (1) cut a cone in the linear space of potential assignments of entropies to regions (entropy space).

The two best-known conditions, which demarcate the holographic entropy cone are:

$$\text{LHS}_{\text{SA}} = S_{A_1} + S_{A_2} \geq S_{A_1 A_2} = \text{RHS}_{\text{SA}}, \tag{3}$$

$$\text{LHS}_{\text{MMI}} = S_{A_1 A_2} + S_{A_2 A_3} + S_{A_3 A_1} \geq S_{A_1} + S_{A_2} + S_{A_3} + S_{A_1 A_2 A_3} = \text{RHS}_{\text{MMI}}. \tag{4}$$

Here 'SA' and 'MMI' stand, respectively, for the subadditivity of entanglement entropy and for the monogamy of mutual information. The former is a consequence of quantum mechanics alone [10] whereas the latter is a strictly holographic condition, which is implied by the Ryu-Takayanagi proposal [8]. With reference to (1) and (2), we may associate $i = 1 \equiv \text{SA}$ and $i = 2 \equiv \text{MMI}$.

As stated, inequalities (1) might include some redundant conditions, which are implied by others. For example, one could combine (3) and (4) and list

$$S_{A_2 A_3} + S_{A_3 A_1} \geq S_{A_3} + S_{A_1 A_2 A_3}, \tag{5}$$

as an additional yet redundant criterion. Inequality (5) is the strong subadditivity of entanglement entropy; like subadditivity, it is a general property of quantum mechanical states [11].

In the holographic context, however, strong subadditivity is superfluous because it is implied by subadditivity and monogamy since $(5) = (3) + (4)$.

One would like to specify the holographic entropy cone without any redundancies—that is, using the smallest possible number of inequalities. Let us assume that the index $i$ in (1) refers only to independent inequalities such as (3) and (4) and so excludes any holographically redundant inequalities like (5). Under the assumption of non-redundancy, the conditions in (1) are facets of the holographic entropy cone. Each facet defines one *independent* way for a state to be marginally consistent with a Ryu-Takayanagi-like interpretation. (Less generic ways to achieve marginality occur on intersections of facets, which form higher codimension loci in entropy space.) In summary:

$$\text{marginally consistent with RT} \quad \Rightarrow \quad \text{LHS}_i = \text{RHS}_i \quad \text{for some } i, \tag{6}$$
$$\text{and} \ \text{LHS}_i > \text{RHS}_i \quad \text{for all others.}$$

Prior to this paper, independent ways to achieve marginality were characterized in [8, 9, 12–14].

**New inequalities**  Recently, reference [15] proved two infinite families of inequalities, which are obeyed by all entropy vectors that are consistent with an RT interpretation. Restated as the contrapositive, the result proven in [15] is:

$$\text{any inequality in [15] is violated} \quad \Rightarrow \quad \text{not consistent with RT.} \tag{7}$$

This assertion has the same structure as (2). In other words, the inequalities in [15] express necessary conditions for a vector of subsystem entropies to be consistent with an RT interpretation.

The two infinite families of inequalities in [15] enjoy a long list of intriguing properties:

- They hold for a topological reason. The topology in question describes nesting relations among subsystems, which are encoded by tessellations of the torus and the projective plane.

- They prove a prior conjecture about the structure of the holographic entropy cone [16] (see also [17]), which was motivated by Page's theorem [18] and unitary models of black hole evaporation [19–23].

- They subsume all but one facet of the holographic entropy cone for mixed states on $N = 5$ regions (the sole exception is inequality number 8 in [12]) as well as the $N = 7$ inequality found in [13].[1]

- They motivate generalizations of the differential entropy formula [24].

It would be nice to add another property to this list:

- (?) They are all facets of the holographic entropy cone.

In other words, we would like to show that the logical statements (6) and (1)—which are conjunctions of conditions labeled by $i$—also include the conditions implied by the inequalities in [15]. This would also mean that the inequalities in [15] cannot be written as combinations of other holographic inequalities the way that strong subadditivity could.

---

[1]All the inequalities found in [15] assume $N$ odd (equivalently, pure states on an even number of parties). Therefore, they have no overlap with the inequalities discovered in [14], which are for mixed states on $N = 6$ regions.

The present paper proves this claim. To contextualize it, we recall that all previously known facets of the holographic entropy cone were shown to be facets individually, not as families.[2] Therefore, we are establishing here the first two known infinite families of facets of the holographic entropy cone.

**Method**   When working with states on $N$ disjoint atomic subsystems $A_1, A_2, \ldots A_N$, we must consider entropies of all non-empty collections of subsystems, for example $A_1A_4$ or $A_2A_5A_6$. This **entropy space** is $d = (2^N - 1)$-dimensional; the $-1$ excludes the empty set.

Consider an inequality LHS $\geq$ RHS, for which implication (7) holds. This means that the entire holographic entropy cone lives in the closed half-space LHS $\geq$ RHS, and has no overlap with the open half-space LHS < RHS. Assume the inequality involves $N$ atomic subsystems and applies to any state, pure or mixed. To prove the 'facetness' of this inequality, it suffices to find $d - 1$ entropy vectors, which:

(i)  saturate the inequality: LHS = RHS;

(ii)  are contained in the holographic entropy cone;

(iii)  are linearly independent.

These conditions mean that the hyperplane of saturation LHS = RHS intersects the holographic entropy cone on a locus of maximal dimension, which is $d - 1$. Given that the entire cone resides on one side of the hyperplane, a max-dimensional overlap with the cone is only possible if the hyperplane is a facet.

We prove that each inequality in [15] is a facet by finding $d - 1 = 2^N - 2$ linearly independent vectors, which are contained in the holographic entropy cone and which saturate our inequalities.

**Organization**   A necessary review and a few useful facts are collected in Section 2. This includes the statement of the inequalities, which we prove to be facets of the holographic entropy cone; they are given in (10) and (11). The proofs of their 'facetness' are given in Sections 3 and 4. We close with a discussion, which sketches some heuristics about the inequalities and their proofs. The appendix contains proofs of some technical lemmas.

## 2   Preliminaries

**Notation**   Our inequalities are most conveniently described in terms of **pure states** on $m + n$ disjoint regions, which we denote $A_i$ ($1 \leq i \leq m$) and $B_j$ (with $1 \leq j \leq n$). This is equivalent to discussing arbitrary (generically, mixed) states on $N = m + n - 1$ disjoint regions because we can always designate one region to be the purifier $O$ and trade all terms containing $O$ for their complements.

The indices on the $A_i$ and $B_j$ are understood periodically (mod $m$) and (mod $n$), for example $A_{m+1} \equiv A_1$. The distinction between $A$- and $B$-type regions, as well as their indexing, is arbitrary. This structure reflects the symmetry of the inequalities, which is $D_m \times D_n$ for the toric inequalities and $D_{2m}$ for the projective plane inequalities. In particular, our notation does not assume or imply any symmetry of the quantum states.

To streamline the discussion, we introduce a special notation for consecutively indexed $k$-tuples of $A$- and $B$-regions:

$$A_i^{(k)} = A_i A_{i+1} \ldots A_{i+k-1}, \qquad \text{and} \qquad B_j^{(k)} = B_j B_{j+1} \ldots B_{j+k-1}. \tag{8}$$

---

[2]The $(m, 1)$ subfamily of toric inequalities (see (10) below), which is usually referred to as the dihedral or cyclic family, was known to be valid [9] but had only been verified as facets for $m = 3$ [8] and $m = 5$ [9].

For $m$ and $n$ odd, it is further useful to distinguish largest consecutively indexed minorities and smallest consecutively indexed majorities of $A$- and $B$-type regions:

$$A_i^{\pm} \equiv A_i^{((m\pm 1)/2)}, \qquad \text{and} \qquad B_j^{\pm} \equiv B_j^{((n\pm 1)/2)}. \qquad (9)$$

## 2.1 The inequalities

We discuss two infinite families of inequalities:

- **Toric inequalities**, which are parametrized by $m, n$ both odd:

$$\sum_{i=1}^{m} \sum_{j=1}^{n} S_{A_i^+ B_j^-} \geq \sum_{i=1}^{m} \sum_{j=1}^{n} S_{A_i^- B_j^-} + S_{A_1 A_2 \dots A_m}. \qquad (10)$$

  We exclude the special case $m = n = 1$, which reads $S_{A_1} \geq S_{A_1}$. Exchanging $m \leftrightarrow n$ merely relabels regions without changing the structure of the inequality. In what follows, without loss of generality, we assume $m \geq n$.

  These inequalities are invariant under $D_m \times D_n$, which acts on regions $A_i$ (respectively $B_j$) like it does on vertices of a regular $m$-gon (respectively $n$-gon).

- **Projective plane inequalities**, which are parametrized by integer $m = n$ of either parity:

$$\frac{1}{2} \sum_{j=1}^{m-1} \sum_{i=1}^{m} \left( S_{A_i^{(j)} B_{i+j}^{(m-j)}} + S_{A_i^{(j)} B_{i+j+1}^{(m-j)}} \right) + (m-1) S_{A_1 A_2 \dots A_m} \geq \sum_{i,j=1}^{m} S_{A_i^{(j-1)} B_{i+j}^{(m-j)}}. \qquad (11)$$

  We exclude the special case $m = 1$, which reads $0 \geq 0$.

  These inequalities are invariant under $D_{2m}$. Its action on the subsystems can be visualized by placing them on vertices of a regular $(2m)$-gon in the cyclic order $\{B_1, A_1, B_2, A_2 \dots B_m, A_m\}$.

A few of these inequalities are already known to be facets of the holographic entropy cone. This includes the toric inequalities with $(m, n)$ equal to $(3, 1)$ [8], $(5, 1)$ [9], $(3, 3)$ [12] and $(5, 3)$ [13], as well as the projective plane inequalities with $m = 2$ [8] and $m = 3$ [12]. Here we prove that all inequalities (10) and (11) are facets.

## 2.2 Useful facts and definitions

**Which entropy vectors are contained in the holographic entropy cone** To prove that (10) and (11) are facets, we find $2^{m+n-1} - 2$ saturating entropy vectors, which are contained in the holographic entropy cone. Let us first review how one verifies that an entropy vector is contained in the holographic entropy cone.

In principle, given a vector of entropies $S_X$ (with $X$ ranging over non-empty collections of fundamental regions $A_i$, $B_j$), one should find a spatial geometry whose minimal surfaces have areas $S_X$. Reference [9] proved that it is enough to find a weighted graph such that:

- a subset of its nodes is labeled $A_i$, $B_j$,

- the minimal cut through the graph, which separates the collection $X$ from its complement, has combined weight $S_X$ (for all $X$).

A few examples of such graphs are shown in Figures 1, 2, 3.

One can think of the graph as a tensor network, which satisfies the minimal cut prescription for computing entanglement entropies. For example, in random tensor network (RTN) states [25] at large bond dimension subsystem entropies are accurately computed by the minimal curt prescription. Therefore, holographic entropy inequalities apply to all RTN states at large bond dimension—which means that they can detect non-randomness in the tensors. This is why we stated in the Introduction that quantum states outside the holographic entropy cone are non-generic in Hilbert space.

In practice, our proof only involves weighted star graphs with one interior central node. Star graphs are combinatorially manageable because the minimal cut computation of $S_X$ has only two cuts to compare: with the central node on the $X$-side or on the $\bar{X}$-side of the cut. To state this more explicitly, let $a_i$ be the weight of the edge that connects the central node to node $A_i$; set a similar definition for $b_j$. (If $A_i$ is absent from the star graph, we let $a_i = 0$.) Let $\mathcal{A}$ be some subset of indices $1 \le i \le m$; and similarly for $\mathcal{B}$. For a region $X$ defined by

$$X = \left(\cup_{i \in \mathcal{A}} A_i\right) \cup \left(\cup_{j \in \mathcal{B}} B_j\right), \tag{12}$$

we have:

$$S_X = \min \left\{ \sum_{i \in \mathcal{A}} a_i + \sum_{j \in \mathcal{B}} b_j, \ \sum_{i \notin \mathcal{A}} a_i + \sum_{j \notin \mathcal{B}} b_j \right\}. \tag{13}$$

**K-basis**  We prove that inequalities (10) and (11) are facets of the holographic entropy cone by finding $2^{m+n-1} - 2$ saturating entropy vectors, which are linearly independent. In verifying linear independence we make ample use of the results of [26].

Reference [26] constructed a convenient basis of entropy vectors. Each basis vector has entropies $S_X$, which are given by minimal cuts on a star graph with an **even** number of arms of weight 1; see Figure 1. We call the number of arms in the star graph its **membership**. The **members** of the star are labeled by the regions $A_i$ and $B_j$, including the region that may be designated as the purifier.

The basis constructed in this way, called K-basis, is complete and not redundant. 'Not redundant' means that all K-basis vectors are linearly independent in entropy space. Completeness is established by counting K-vectors of all memberships:

$$\sum_{p=1}^{\lfloor (N+1)/2 \rfloor} \binom{N+1}{2p} = 2^N - 1. \tag{14}$$

In what follows, K-basis vectors with membership $2p$ will be called **K(2p)-vectors**, for example K2-vectors, K4-vectors etc.

The preceding discussion implies that K-vectors have the following useful properties:

**Fact 1** *K-vectors satisfy every inequality (10-11) because they are, from their definition, contained in the holographic entropy cone.*

**Fact 2** *Consider an arbitrarily weighted star graph and collect the regions with non-vanishing bond weights into a set $\mathcal{S}$. (In other words, $\mathcal{S}$ is the set of members of the star.) When we decompose the entropy vector of the star graph into K-basis components, only K-vectors with members in $\mathcal{S}$ have non-vanishing coefficients. This follows from completeness in the entropy space for $\mathcal{S}$.*

Let us briefly illustrate Fact 2. Denote entropy vectors of weighted star graphs with $\vec{s}(\text{region}^{\text{weight}}, \ldots)$. One example of a K-basis decomposition is:

$$\vec{s}(X_0^2, X_1^1, X_2^1, X_3^1, X_4^0) = \tfrac{1}{2}\vec{s}(X_0^1, X_1^1, X_2^1, X_3^1, X_4^0) + \tfrac{1}{2}\vec{s}(X_0^1, X_1^1, X_2^0, X_3^0, X_4^0)$$
$$+ \tfrac{1}{2}\vec{s}(X_0^1, X_1^0, X_2^1, X_3^0, X_4^0) + \tfrac{1}{2}\vec{s}(X_0^1, X_1^0, X_2^0, X_3^1, X_4^0). \tag{15}$$

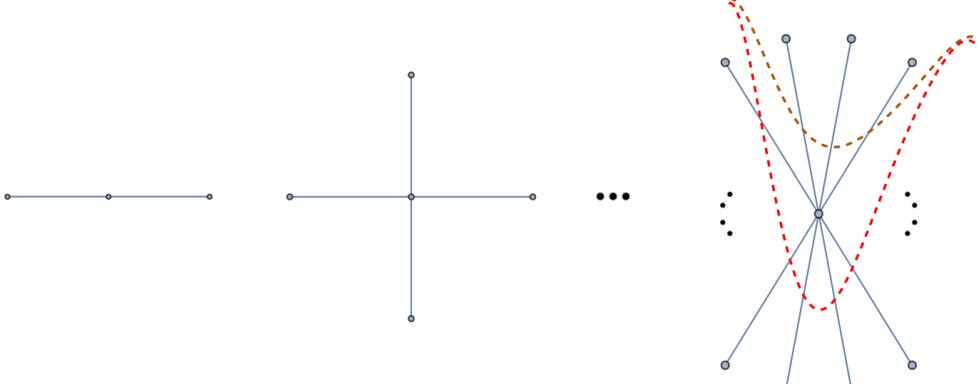

Figure 1: Star graphs, which define the K-basis [26], have an even number of external vertices. The vertices are labeled by subsystems $A_i$ and $B_j$ (which we call members) and all bonds have weight 1. In the rightmost panel, we show two candidate minimal cuts for a region with three members as an illustration of equation (13).

Fact 2 says that because region $X_4$ has zero weight on the left hand side (it is not a member of the star), it also has zero weight in all components on the right hand side. The terms on the right hand side of (15) are K-vectors because each contains an even number of weights 1 and all others are zero.

From now on, we will drop weight-0 regions from expressions $\vec{s}(\text{region}^{\text{weight}}, \dots)$.

**Evaluating the inequality on a star graph**    Let us temporarily write inequalities (10) and (11) in the form LHS − RHS ≥ 0. Quantity LHS − RHS sends an entropy vector to a number. With a slight abuse of language, we call this number **the value of the inequality on the graph** or on its corresponding entropy vector.

## 3 Toric inequalities are facets – The proof

The lemmas in this section, which are not demonstrated in the main text, are proven in the appendix.

In general terms, our strategy is to find a special family of star graphs such that:

(i)  They saturate the toric inequalities (10). In other words, inequalities (10) evaluate to 0 on these graphs.

(ii)  The star graphs are in one-to-one correspondence with K-basis vectors, such that their linear independence is manifest.

If one can find such a family of $d - 1 = 2^N - 2$ vectors (where $N = n + m - 1$) then the proof is complete. As stated, condition (ii) above is not precise. That is because the saturating star graphs relate to K-vectors in several different ways. We discuss them below in separate subsections.

Before that, we make one simple but useful observation:

**Fact 3** *Consider two K-vectors whose members are related by a $D_m$ transformation on the A-members and a $D_n$ transformation on the B-members. (For example, take the pair of K4-vectors with members $\{A_1, B_1, B_2, B_4\}$ and $\{A_3, B_2, B_3, B_5\}$.) Evaluating inequalities (10) on the pair gives the same number.*

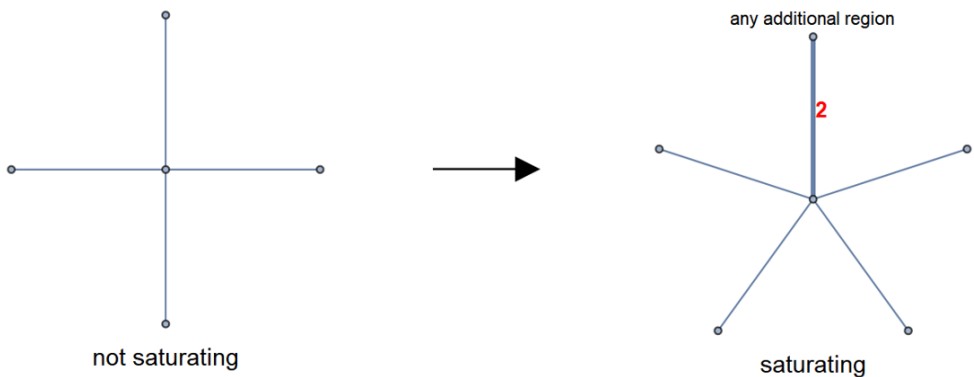

Figure 2: An illustration of Lemma 3 and Lemma 14. Unmarked bonds have weight 1.

This holds because inequalities (10) are invariant under $D_m \times D_n$.

Stated differently, Fact 3 says that K-vectors are naturally organized into multiplets of the symmetry $D_m \times D_n$. Evaluating the inequality on a K-vector is an invariant of each multiplet.

## 3.1 EPR pairs

EPR pairs are K2-vectors. All of them individually saturate the toric inequalities. This property of the toric inequalities is called superbalance [27].

For easy reference in the future, we highlight again: K2-vectors provide

$$\#_2 = \binom{N+1}{2} \tag{16}$$

linearly independent entropy vectors, which saturate inequalities (10).

## 3.2 K4-vectors

The analysis of K4-vectors is somewhat intricate. We organize it into a series of lemmas. Lemmas 1-3 are proved in the Appendix.

**Lemma 1** *Every toric inequality evaluated on a K4-vector gives either 0 or 2.*

**Lemma 2** *If a K4-vector does not saturate inequality (10) then its members include one A-type region and three B-type regions, or vice versa.*

Based on the lemma, we henceforth distinguish **non-saturating AAAB-vectors** and **non-saturating ABBB-vectors**.

**Lemma 3** *Suppose a K4-vector does not saturate some inequality (10). Take any non-member of that K4-vector and adjoin it with bond weight 2, so as to form a star graph with five arms with weights $\{2, 1, 1, 1, 1\}$. Then the resulting star graph saturates (10).*

Lemma 3 is illustrated in Figure 2.

Next, we need the K-vector decomposition of the five-armed star graph with weights $\{2, 1, 1, 1, 1\}$:

**Lemma 4** *In the notation of equation (15), we have:*

$$\vec{s}(X_0^2, X_1^1, X_2^1, X_3^1, X_4^1) = \tfrac{1}{2}\vec{s}(X_0^1, X_1^1, X_2^1, X_3^1) + \tfrac{1}{2}\vec{s}(X_0^1, X_1^1, X_2^1, X_4^1)$$
$$+ \tfrac{1}{2}\vec{s}(X_0^1, X_1^1, X_3^1, X_4^1) + \tfrac{1}{2}\vec{s}(X_0^1, X_2^1, X_3^1, X_4^1)$$
$$- \tfrac{1}{2}\vec{s}(X_1^1, X_2^1, X_3^1, X_4^1). \tag{17}$$

This is easily confirmed by a direct calculation.

We now split the star graphs stipulated in Lemma 3 into two classes. Based on Lemma 2, the members of such a star graph are:

(1) either two *A*-regions and three *B*-regions (or vice versa)

(2) or one *A*-region and four *B*-regions (or vice versa).

Let us apply Lemma 4 to both cases. Without loss of generality, we adjoin a fifth region to a non-saturating ABBB-vector $\vec{s}(A_i^1, B_{j_1}^1, B_{j_2}^1, B_{j_3}^1)$. We mark the fifth, weight-2 region with a $*$ in the subscript. We then find:

Case (1): $\quad \vec{s}(A_*^2, A_i^1, B_{j_1}^1, B_{j_2}^1, B_{j_3}^1) = \tfrac{1}{2}\vec{s}(A_*^1, A_i^1, B_{j_1}^1, B_{j_2}^1) + \tfrac{1}{2}\vec{s}(A_*^1, A_i^1, B_{j_1}^1, B_{j_3}^1)$

$$+ \tfrac{1}{2}\vec{s}(A_*^1, A_i^1, B_{j_2}^1, B_{j_3}^1) \boxed{+ \tfrac{1}{2}\vec{s}(A_*^1, B_{j_1}^1, B_{j_2}^1, B_{j_3}^1)}$$

$$- \tfrac{1}{2}\vec{s}(A_i^1, B_{j_1}^1, B_{j_2}^1, B_{j_3}^1), \tag{18}$$

Case (2): $\quad \vec{s}(A_i^1, B_{j_1}^1, B_{j_2}^1, B_{j_3}^1, B_*^2) = \tfrac{1}{2}\vec{s}(B_{j_1}^1, B_{j_2}^1, B_{j_3}^1, B_*^1) \boxed{+ \tfrac{1}{2}\vec{s}(A_i^1, B_{j_2}^1, B_{j_3}^1, B_*^1)}$

$$\boxed{+ \tfrac{1}{2}\vec{s}(A_i^1, B_{j_1}^1, B_{j_3}^1, B_*^1)} \boxed{+ \tfrac{1}{2}\vec{s}(A_i^1, B_{j_1}^1, B_{j_2}^1, B_*^1)}$$

$$- \tfrac{1}{2}\vec{s}(A_i^1, B_{j_1}^1, B_{j_2}^1, B_{j_3}^1). \tag{19}$$

The boxes will be explained momentarily. The goal of this exercise is to derive:

**Lemma 5** *The star graphs stipulated in Lemma 3 obey the linear relations:*

$$\textit{sum of coefficients of non-saturating AAAB-vectors} = 0, \tag{20}$$
$$\textit{sum of coefficients of non-saturating ABBB-vectors} = 0. \tag{21}$$

Because this lemma is conceptually important for the structure of the proof, we verify it in the main text.

Evaluate the inequality on entropy vectors (18) and (19). Because the graphs were obtained from Lemma 3, this gives 0. On the other hand, evaluating the inequality on $-\tfrac{1}{2}\vec{s}(A_i^1, B_{j_1}^1, B_{j_2}^1, B_{j_3}^1)$ gives $-1$ because it is the initial non-saturating K4-vector. By Lemma 1, the terms with positive coefficients in (18) and (19) can only evaluate to 0 or 1. Therefore, precisely one of them must evaluate to $+1$. This selects a second non-vanishing ABBB-vector in the expansion (18, 19).

In Case (1) above, the other non-saturating ABBB-vector in expansion (18) is highlighted in the box. We know it must be non-saturating because the others are of the AABB type, which always saturates the toric inequalities (Lemma 2). In Case (2), on the other hand, we have three candidates for which component is non-saturating; we highlighted them in the boxes in equation (19). Precisely one of them is non-saturating.

In all cases, the K-vector expansion of the star graph from Lemma 3 contains precisely two non-saturating K-vectors: the initial non-saturating K4-vector assumed in Lemma 3, and one other one. Their coefficients in the expansion are $-1/2$ and $+1/2$, respectively. Crucially, if the initial non-saturating K4-vector is of ABBB type then so is the other non-saturating vector in the expansion; this is clear from equations (18) and (19). (The analysis is identical if we start from a non-saturating AAAB-vector.) This establishes Lemma 5.

**Saturating entropy vectors built from K4-vectors**    Consider inequality (10) at some fixed $(m, n)$. Let us divide the linear span of all K4-vectors in entropy space into three components:

- The linear span of those K4-vectors, which saturate the inequality; call it $V_0$.

- The linear span of non-saturating AAAB-vectors; call it $V_{AAAB}$.

- The linear span of non-saturating ABBB-vectors; call it $V_{ABBB}$. We have assumed $m \geq n$ and excluded $(m, n) = (1, 1)$ so $m \geq 3$. This leaves out the special possibility $n = 1$, which refers to the dihedral inequalities proven in [9]. When $n = 1$, the space $V_{ABBB}$ is empty.

By Lemma 2, the direct sum $V_0 \oplus V_{AAAB} \oplus V_{ABBB}$ equals the linear span of the K4-vectors.

Lemmas 1 and 3 construct for us a large family of entropy vectors, which live in the holographic entropy cone and which saturate the $(m, n)$ toric inequality:

- The K4-vectors, which saturate the inequality. By definition, they span $V_0$.

- The star graphs from Lemma 3, which originate from non-saturating AAAB-vectors. Equations (18, 19) show that they live in $V_{AAAB} \oplus V_0$.

- The star graphs from Lemma 3, which originate from non-saturating ABBB-vectors. They live in $V_{ABBB} \oplus V_0$. Again, if $n = 1$ then this set is empty.

Assuming $n \geq 3$, Lemma 5 says that all these vectors live in a codimension-2 subspace of the span of K4-vectors. It is the codimension-2 subspace described by equations (20-21). If $n = 1$ then these vectors live in a (codimension-1) hyperplane described by equation (20). This is because equation (21) does not impose a constraint: the coefficients of saturating ABBB-vectors trivially add up to zero simply because such vectors do not exist.

Our next task is to show that these vectors do in fact span the entire codimension-2 subspace (respectively codimension-1 hyperplane if $n = 1$) in the linear space generated by K4-vectors. In other words:

**Lemma 6** *There are no additional linear dependencies among the inequality-saturating entropy vectors from Lemmas 1 and 3, other than equations (20) and (21).*

The only proof we found is a little clanky; see Appendix. Lemma 6 is equivalent to the following statement:

**Lemma 7** *For $n \geq 3$, there exists a basis for the linear span of K4-vectors of the form:*

$$
\begin{aligned}
\vec{k}_{AAAB} &= \text{the average of all non-saturating AAAB-vectors}, \\
\vec{k}_{ABBB} &= \text{the average of all non-saturating ABBB-vectors}, \\
\vec{k}_i \quad &\text{(where each } \vec{k}_i \text{ saturates the inequality)}.
\end{aligned}
\tag{22}
$$

Even though Lemma 7 is essentially synonymous with Lemma 6, we list the latter separately because it will be handy in the next subsection.

**Summary**    For any fixed $(m, n)$, we have constructed a collection of entropy vectors, which:

- are contained in the holographic entropy cone,

- saturate the $(m, n)$ toric inequality,

- live in the linear span of K4-vectors,

- span a subspace in that vector space, which is codimension-2 (for $n \geq 3$) or codimension-1 (for $n = 1$).

In this way, K4-vectors provide

$$\#_4 = \begin{cases} \binom{N+1}{4} - 2 & (n \geq 3), \\ \binom{N+1}{4} - 1 & (n = 1), \end{cases} \tag{23}$$

linearly independent entropy vectors, which saturate inequalities (10).

### 3.3 K6-vectors

We distinguish **saturating** and **non-saturating K6-vectors**, just like we did before for K4-vectors.

We first establish a lemma, which will also be useful for analyzing K-vectors of higher membership. That is why we phrase it in terms of general K($2p$)-vectors.

**Lemma 8** *Consider the toric inequality at some fixed $(m, n)$. If a K($2p$)-vector ($p \geq 3$) does not saturate the inequality then resetting one of its weights to $2p-3$ produces an entropy vector, which does saturate it.*

This result, which is illustrated in Figure 3, is essential for our argument. Unfortunately, we have not been able to find a simple proof of this statement. A valid but complicated proof is given in Appendix A.3.

**Lemma 9** *The K-vector expansion of the graph stipulated in Lemma 8 is:*

$$\vec{s}(X_0^{2p-3}, X_1^1, \dots X_{2p-1}^1) = \sum_{q=2}^{p} \frac{(-1)^q (q-1)!}{\prod_{r=3}^{q+1}(2p-r)} \sum_{1 \leq i_{(\cdot)} \leq 2p-1} \vec{s}(X_0^1, X_{i_1}^1, X_{i_2}^1, \dots X_{i_{2q-1}}^1)$$

$$+ \sum_{q=2}^{p-1} \frac{(-1)^{q-1}(q-1)(q-1)!}{\prod_{r=3}^{q+2}(2p-r)} \sum_{1 \leq i_{(\cdot)} \leq 2p-1} \vec{s}(X_{i_1}^1, X_{i_2}^1, \dots X_{i_{2q}}^1). \tag{24}$$

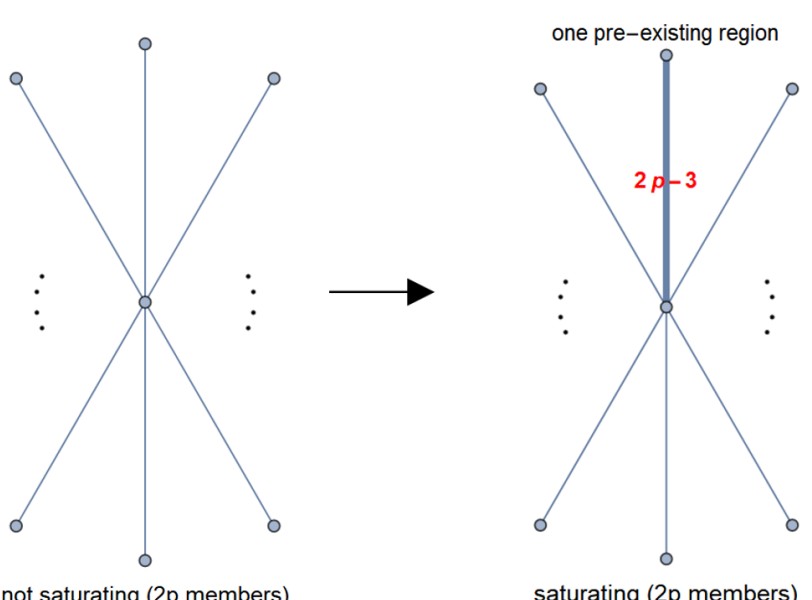

Figure 3: An illustration of Lemma 8 and Lemma 16. Unmarked bonds have weight 1.

This equation can be verified by a direct calculation. The sums $\sum_{i_{(\cdot)}}$ are over **distinct** $(2q-1)$- and $(2q)$-tuples, which mark the members of the K-vectors in the expansion. For our purposes, the important fact is that expansion (24) contains a unique K$(2p)$-vector with a non-vanishing coefficient:

$$\text{coefficient of the unique K}(2p)\text{-vector in expansion (24)} = (-1)^p \bigg/ \binom{2p-3}{p-1}. \qquad (25)$$

Applied at $p = 3$, Lemma 9 tells us that the vectors constructed in Lemma 8 saturate the inequalities by balancing the value of the inequality on the initial K6-vector against its values on K4-vectors. In particular, a subset of the K4-vectors that appear in (24) at $p = 3$ must be non-saturating. Non-saturating K4-vectors were sorted in Lemma 2 into two categories: non-saturating AAAB-vectors and non-saturating ABBB-vectors. The next question we must answer is:

- Does the expansion in Lemma 9 (for $p = 3$) contain both classes of non-saturating K4-vectors or only one class?

The members of the K4-vectors in the expansion are subsets of the members of the initial non-saturating K6-vector, which is the starting point in Lemma 8. To answer the above question with "both," the six members of the initial K6-vector would have to be of the form AAABBB. That is because only this combination contains both an AAAB and an ABBB subset.

Indeed, when we apply Lemma 8 to a non-saturating AAABBB-vector, expansion (24) does contain non-saturating K4-vectors of both kinds. However, a special circumstance occurs there, which we state as a separate lemma:

**Lemma 10** *Apply Lemma 8 to a non-saturating K6-vector. Then in the expansion (24) one of the following holds:*

- *The coefficients of the non-saturating AAAB-vectors add up to zero.*

- *The coefficients of the non-saturating ABBB-vectors add up to zero.*

*Moreover, which circumstance occurs depends on whether we set to 3 the weight of an A-bond or a B-bond.*

We prove this in the Appendix.

We now combine Lemma 10 with Lemma 7. The latter says that expansion (24) can be written in one of these two forms:

$$\vec{s}(X_0^3, X_1^1, X_2^1, X_3^1, X_4^1, X_5^1) = -\frac{1}{3}\vec{s}(X_0^1, X_1^1, X_2^1, X_3^1, X_4^1, X_5^1) + c\,\vec{k}_{AAAB} + \text{saturating}, \qquad (26)$$

or:

$$\vec{s}(X_0^3, X_1^1, X_2^1, X_3^1, X_4^1, X_5^1) = -\frac{1}{3}\vec{s}(X_0^1, X_1^1, X_2^1, X_3^1, X_4^1, X_5^1) + c\,\vec{k}_{ABBB} + \text{saturating}. \qquad (27)$$

The difference is in the appearance of either $\vec{k}_{AAAB}$ or $\vec{k}_{ABBB}$ but not both.

**Saturating entropy vectors built from K6-vectors** Take inequality (10) at some fixed $(m, n)$. Consider the following collection of vectors:

- The saturating K6-vectors.

- For each non-saturating K6-vector, take one saturating vector, which is constructed in Lemma 8.

By construction, these vectors live in the holographic entropy cone and saturate the inequality. They are clearly linearly independent from each other and from the vector spaces constructed in Subsections 3.1 and 3.2. This is because each of them has a unique non-zero entry in the K6-part of the K-basis expansion, viz. equation (25).

Vectors constructed in this way provide

$$\#_6 = \binom{N+1}{6} \qquad (n=1) \tag{28}$$

linearly independent vectors to the locus of saturation of the $(m, n)$ toric inequality. This is the final result for $n = 1$. At $n \geq 3$, however, we can add one additional saturating vector, which is linearly independent of all the rest.

A key observation is that when $n \geq 3$, there is at least one saturating K6-vector with structure AAABBB, to which Lemma 8 can be applied in more than one way: by setting weight 3 in the star graph either to an A-bond or to a B-bond. We state this assertion as a separate lemma:

**Lemma 11** *When $n \geq 3$, there exists a non-saturating K6-vector, to which Lemma 8 can be applied both on an A-bond and on a B-bond.*

By Lemma 10, the resulting entropy vectors will have expansions (26) and (27).

We now augment our list of saturating entropy vectors by this one additional vector, which is guaranteed by Lemma 11. If the additional vector is linearly independent from the others, we will have constructed

$$\#_6 = \binom{N+1}{6} + 1 \qquad (n \geq 3) \tag{29}$$

linearly independent vectors in the locus of saturation of the $(m, n)$ toric inequality. A meticulous reader might object that linear independence only implies $\#_6 \geq \binom{N+1}{6} + 1$. However, we cannot have $\#_6 > \binom{N+1}{6} + 1$ because otherwise all K4-vectors and all K6-vectors would saturate the inequality.

As a final step in verifying equation (29), we prove:

**Lemma 12** *For $n \geq 3$, take the entropy vectors in the collection:*

- *Saturating K6-vectors.*

- *One vector from Lemma 8, constructed for every non-saturating K6-vector.*

- *For one K6-vector stipulated in Lemma 11, take the other application of Lemma 8, which is not included in the previous item.*

*Then there are no linear dependencies among these $\binom{N+1}{6} + 1$ vectors.*

To prove this, expand these vectors in the basis, which consists of K6-vectors and the K4-basis from Lemma 7. We collect the coefficients of vectors in Lemma 12 into a matrix. The specific values of the entries are immaterial, only their position and their non-vanishing character matters. Therefore, we simply mark non-vanishing entries with 'x'.

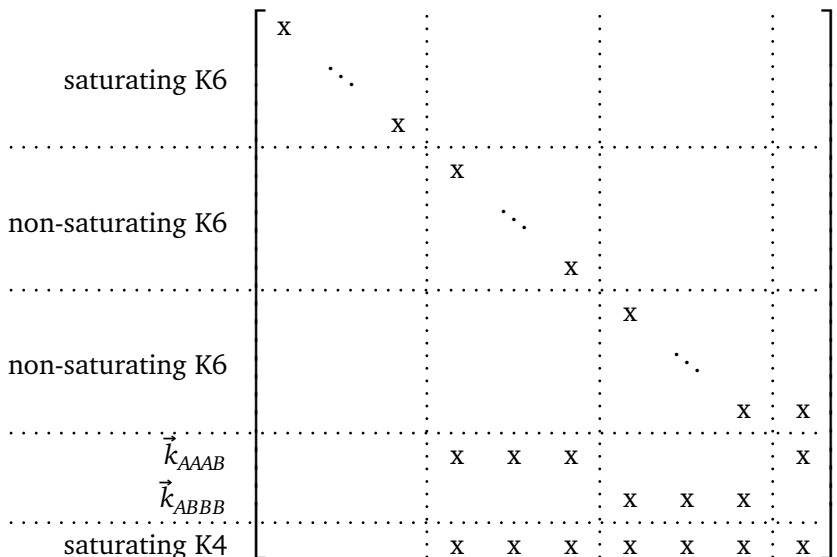

It is clear that this matrix has maximal rank.

## 3.4 K8- through K($n + m$)-vectors

If $m+n \leq 6$ then the preceding sections suffice to prove that the $(m,n)$ toric inequality is a facet of the holographic entropy cone. In any case, the 'facetness' of those specific inequalities— $(m,n) = (3,1),(5,1),(3,3)$—was already proven in [8, 9, 12]. From now on we assume $m + n \geq 8$.

**Saturating entropy vectors built from K6-vectors** Fix $(m,n)$ and $8 \leq 2p \leq m + n$ and consider the following collection of vectors:

- The saturating K($2p$)-vectors.

- For each non-saturating K($2p$)-vector, take one saturating vector, which is constructed in Lemma 8.

By construction, these vectors live in the holographic entropy cone and saturate the inequality. They are clearly linearly independent from each other and from similarly constructed vectors with $p' < p$. This is because each of them has a unique non-zero entry in the K($2p$)-part of the K-basis expansion, as asserted in Lemma 9 and equation (25).

Because they are in one-to-one correspondence with K($2p$)-vectors, we have:

$$\#_{2p} = \binom{N + 1}{2p}. \tag{30}$$

## 3.5 Summary

From Section 3.1 to Section 3.4, we have constructed

$$\sum_{p=1}^{\lfloor (N+1)/2 \rfloor} \#_{2p} = 2^N - 2, \tag{31}$$

linearly independent entropy vectors, which live in the holographic entropy cone and which saturate the $(m,n)$ toric inequality. This completes the proof.

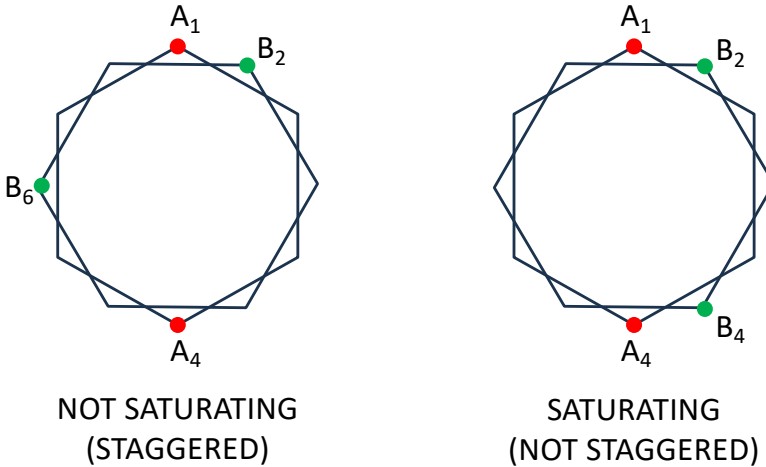

Figure 4: An illustration of Lemma 13.

## 4 Projective plane inequalities are facets – The proof

There are $m$ regions $A_i$ and $m$ regions $B_j$, so the dimension of entropy space is $2^{2m-1} - 1$. The analysis is similar to Section 3 but significantly simpler.

### 4.1 EPR pairs

All EPR pairs saturate the projective plane inequalities because they are superbalanced [27]:

$$\#_2 = \binom{2m}{2}. \tag{32}$$

### 4.2 K4-vectors

The analysis is easier than for the toric inequalities because there are no 'selection sectors,' which would be analogous to non-saturating AAAB-vectors and non-saturating ABBB-vectors. For the projective plane inequalities, all non-saturating K4-vectors live in one family and we can achieve saturation by canceling them off one another in arbitrary pairs. We state this in the form of three lemmas, which are proved in Appendix A.3:

**Lemma 13** *A K4-vector does not saturate inequality (11) if and only if the following two conditions are met:*

- *Its members include two A-type regions and two B-type regions.*

- *Call the two A-members $A_{i_1}$ and $A_{i_2}$ with $1 \leq i_1 < i_2 \leq m$ and call the two B-members $B_{j_1}$ and $B_{j_2}$, where the indices (which are valued (mod $m$)) are taken from $i_1 + 1 \leq j_1 < j_2 \leq i_1 + m$. With these conventions, the condition is:*

$$i_1 + 1 \leq j_1 \leq i_2, \qquad \text{and} \qquad i_2 + 1 \leq j_2 \leq i_1 + m. \tag{33}$$

The lemma is illustrated in Figure 4. Note that condition (33) can equally well be phrased as a restriction on the $i_{1,2}$ indices relative to the $j_{1,2}$ indices:

$$j_1 \leq i_2 \leq j_2 - 1, \qquad \text{and} \qquad j_2 \leq i_1 + m \leq j_1 - 1 + m. \tag{34}$$

Its form is not identical to (33), but it becomes identical after a simple relabeling: $j_{1,2} \to j_{1,2}+1$ and $i_1 \leftrightarrow i_2$. Therefore, in proving lemmas, any reasoning about $A$-regions will also apply to $B$-regions, and vice versa.

**Lemma 14** *Suppose a K4-vector does not saturate some inequality (11). Take any non-member of that K4-vector and adjoin it with bond weight 2, so as to form a star graph with five arms with weights $\{2,1,1,1,1\}$. Then the resulting star graph saturates the inequality.*

Lemma 14 is illustrated in Figure 2.

**Lemma 15** *Working at fixed m, call $d_*$ the number of K4-vectors, which do not saturate inequality (11). Then the vectors obtained in Lemma 14 span a linear space of dimension $d_* - 1$.*

**Summary** For any fixed $m$, K4-vectors which individually saturate inequality (11) and vectors from Lemma 14 together give us

$$\#_4 = \binom{2m}{4} - 1 \tag{35}$$

linearly independent vectors. All of them are contained in the holographic entropy cone.

### 4.3 K6- through K(2$m$)-vectors

**Lemma 16** *Consider the projective plane inequality at some fixed m. If a K(2p)-vector ($p \geq 3$) does not saturate the inequality then resetting one of its weights to $2p - 3$ produces an entropy vector, which does saturate it.*

The lemma is illustrated in Figure 3 and proven in Appendix A.3.

Fix $m$ and $3 \leq p \leq m$ and consider the following collection of vectors:

- The K(2$p$)-vectors, which saturate the projective plane inequality at $m$.

- For each non-saturating K(2$p$)-vector, take one saturating vector, which is constructed in Lemma 8.

By construction, these vectors live in the holographic entropy cone and saturate the inequality. They are clearly linearly independent from each other and from similarly constructed vectors with $p' < p$. This is because each of them has a unique non-zero entry in the K(2$p$)-part of the K-basis expansion, as asserted in Lemma 9 and equation (25).

Because they are in one-to-one correspondence with K(2$p$)-vectors, we have:

$$\#_{2p} = \binom{2m}{2p}. \tag{36}$$

### 4.4 Summary

From Section 4.1 to Section 4.3, we have constructed

$$\sum_{p=1}^{m} \#_{2p} = 2^{2m-1} - 2 \tag{37}$$

linearly independent entropy vectors, which live in the holographic entropy cone and which saturate the projective plane inequality $m$. This completes the proof.

# 5  Discussion

It is striking that star graphs are enough to saturate inequalities (10) and (11) in every linearly independent way. What is more, a very restricted set of star graphs suffices:

- K-basis vectors, see Figure 1,

- star graphs with one unequal weight, so-called flower graphs of [16], see Figure 2 and Figure 3.

What controls the saturation of the toric inequalities is the distribution of the members of these star graphs around the $m$-gon of the $A_i$ subsystems (and the $n$-gon of the $B_j$ subsystems). The ordering of the regions in the polygons is set by the inequality in question.

Roughly, the toric inequalities applied to star graphs quantify how evenly the members of the star are distributed on the polygons. This means that a highly uneven selection of $A$-members and $B$-members of the K-vector saturates the inequality. On the other hand, a K-vector whose $A$-members and $B$-members have indices that are relatively evenly distributed around $\mathbb{Z}_m$ and $\mathbb{Z}_n$ is far from saturation. These statements are quantified in the Appendix, their most explicit form being equation (A.16).

In this way, the toric inequalities encapsulate a kind of interplay between the entanglement structures on the $A$- and $B$-regions, which is organized by dihedral symmetry. By analyzing the inequalities in the K-basis, we see that this interplay plays out primarily at the level of K4-vectors, see Lemma 5 and equation (23). In particular, the locus of saturation of the inequality is codimension-2 with respect to the linear span of K4-vectors, but co-dimension-1 once K6-vectors are included. This suggests that the toric inequalities might have a heuristic interpretation at the level of four-party versus six-party entanglement. It would be clarifying to find such an interpretation.

As for the projective plane inequalities, applying them to star graphs also measures how evenly the members of the star are distributed on the $m$-gons of regions $A_i$ and $B_j$. Unlike the toric inequalities, however, here the unevenness is measured in relative terms: $A$-regions relative to $B$-regions. This is especially evident with K4-vectors, which do not saturate inequalities (11) if and only if their $A$-members and $B$-members have staggered indices; see Figure 4 and Lemma 13 for the quantitative statement. The behavior of higher K($2p$)-vectors under the projective plane inequalities, which is studied in the proof of Lemma 16 and illustrated in Figure 8, invokes similar heuristics.

Inequalities (10) and (11) were first identified in an effort to prove the conjectured form of the holographic cone of *average* entropies [16], which in turn was motivated by an analysis of black hole evaporation [19, 20] and Page's theorem [18]. In the latter context, star graphs with one distinct weight model old black holes at various stages of evaporation, the many uniformly weighted legs of a star graph corresponding to the many components of previously emitted Hawking radiation. Under this identification, we have shown that evaporating old black holes saturate all the toric and projective plane inequalities. The qualitative meaning of the inequalities—capturing how concentrated / diluted information is on an imaginary $m$-gon and $n$-gon of subsystems—also carries over to the black hole context. Recast in that language, the toric inequalities are unsaturated only if two particles of Hawking radiation develop a non-vanishing mutual information, conditioned on half of the previously emitted radiation; see Lemma 17 for the quantitative statement. A similar heuristic interpretation applies to the projective plane inequalities.



# Acknowledgments

BC thanks Sirui Shuai, Yixu Wang and Daiming Zhang for collaboration on [15], which proved inequalities (10) and (11). We thank Matthew Headrick, Sergio Hernández Cuenca, Veronika Hubeny, Mukund Rangamani, Sirui Shuai and Yixu Wang for discussions, and further thank Sirui Shuai for sharing some handy Mathematica code. BC thanks for hospitality the Tsinghua Southeast Asia Center on Kura Kura Bali, Indonesia where this work was completed. This manuscript was typeset using the publicly available JHEP template.

**Funding information** This work was supported by an NSFC grant number 12342501, the BJNSF grant 'Holographic Entropy Cone and Applications,' and a Dushi Zhuanxiang Fellowship.

# A    Proofs of lemmas

## A.1    Preliminaries

The following rewriting of the toric inequalities (10) is useful:

$$\frac{1}{2}\sum_{i=1}^{m}\sum_{j=1}^{n}I(A_i:B_j\,|\,A_{i+1}^-B_{j+1}^-)\geq S_{A_1A_2...A_m}\,. \tag{A.1}$$

Here $I(X:Y|Z)$ is the conditional mutual information, $S_{XZ}+S_{YZ}-S_Z-S_{XYZ}$. The rewriting follows from adding up two copies of (10), reindexed and with some terms traded for their complements:

$$\sum_{i=1}^{m}\sum_{j=1}^{n}S_{A_i^+B_{j+1}^-}\geq\sum_{i=1}^{m}\sum_{j=1}^{n}S_{A_{i+1}^-B_{j+1}^-}+S_{A_1A_2...A_m}\,, \tag{A.2}$$

$$\sum_{i=1}^{m}\sum_{j=1}^{n}S_{A_{i+1}^-B_j^+}\geq\sum_{i=1}^{m}\sum_{j=1}^{n}S_{A_i^+B_j^+}+S_{A_1A_2...A_m}\,. \tag{A.3}$$

Rewriting (A.1) is helpful for visualizing which K($2p$)-vectors, and why, do not saturate an $(m,n)$ toric inequality. We develop it presently.

In a K($2p$)-vector, a pair of single members have non-vanishing conditional mutual information if and only if it is conditioned on a set containing $p-1$ members. If such a conditional mutual information is non-vanishing, it equals 2. Therefore, we can rewrite the value of the inequality on a K($2p$)-vector in the following way:

$$\#\{\text{pairs }(A_i,B_j)\text{ of K}(2p)\text{-members s.t. }A_{i+1}^-B_{j+1}^-\text{ contains }p-1\text{ members}\}-S_{A_1A_2...A_m}\,. \tag{A.4}$$

To keep track of how many K-vector members are in $A_{i+1}^-B_{j+1}^-$, we introduce two functions. These functions implicitly depend on the K($2p$)-vector in question, but we will not write that dependence explicitly.

$$\tilde{g}(i)=\#\{A_j\text{ s.t. }i+1\leq j\leq i+(m-1)/2\text{ and }A_j\text{ is a member of the K}(2p)\text{-vector}\}\,, \tag{A.5}$$

$$\tilde{h}(i)=\#\{B_j\text{ s.t. }i+1\leq j\leq i+(n-1)/2\text{ and }B_j\text{ is a member of the K}(2p)\text{-vector}\}\,. \tag{A.6}$$

For easy reference in the future, we write explicitly the value of the inequality on a K($2p$)-vector:

$$\#\{\text{pairs }(i,j)\text{ s.t. }A_i\text{ and }B_j\text{ are members and }\tilde{g}(i)+\tilde{h}(j)=p-1\}-S_{A_1A_2...A_m}\,. \tag{A.7}$$

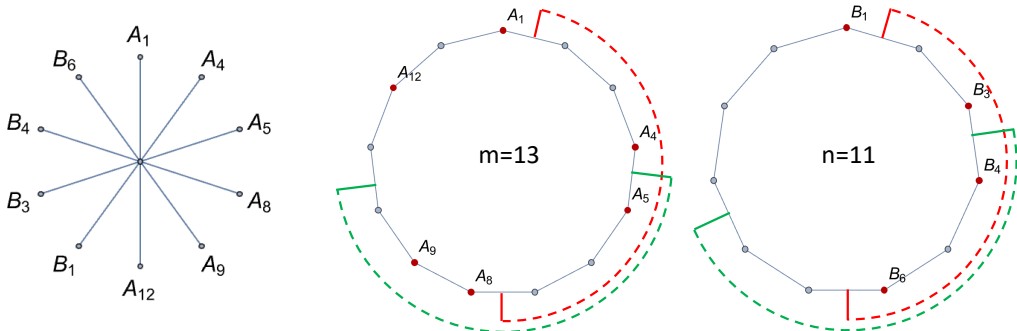

Figure 5: A K10-vector with six $A$-members ($k = 6$) and four $B$-members ($l = 4$). The function $g(t)$ counts other $A$-members in the 'almost-semicircle' clockwise from the $t^{\text{th}}$ $A$-member. In the middle panel, which displays $A$-regions, we mark the 'almost-semicircles' for $g(1) = 2$ (red) and $g(2) = 3$ (green). In the right panel, which displays $B$-regions, we mark the 'almost-semicircles' for $h(1) = 3$ (red) and $h(2) = 2$ (green).

Equation (A.7) only involves the values of $\tilde{g}$ and $\tilde{h}$ on members of the K($2p$)-vector. Once $\tilde{g}$ and $\tilde{h}$ are known on the members, we can forget about the specific indices $i$ and $j$ of those members. For this reason, we introduce new functions $g$ and $h$, which remove the information about non-members:

$$g(t) = \tilde{g}(\text{subscript of the } t^{\text{th}} A\text{-member of the K}(2p)\text{-vector}), \qquad (A.8)$$

$$h(t) = \tilde{h}(\text{subscript of the } t^{\text{th}} B\text{-member of the K}(2p)\text{-vector}). \qquad (A.9)$$

Equivalently, if we label the $A$-members of the K-vector $A_{i_t}$ (with $1 \le t \le k$ where $k$ counts $A$-members of the K-vector) then $g(t) = \tilde{g}(i_t)$. Function $g(t)$ looks at consecutive $A$-members of the K-vector and, for each of them, counts other $A$-members that reside in the 'almost-semicircle' right next to it. We say 'almost-semicircle' because we have $m$ $A$-subsystems and we are counting members in a range of $(m-1)/2$ of them.

The definition of $g(t)$ and our other conventions are illustrated in Figure 5.

**Examples** Suppose that $m = 13$ and consider a K($2p$)-vector, which has six $A$-members: $\{A_1, A_4, A_5, A_8, A_9, A_{12}\}$; see the middle panel in Figure 5. The regions $A_{i+1}^{-}$ have $(m-1)/2 = 6$ elements. Then:

$$
\begin{aligned}
g(1) &= 2, & \text{because } & A_4, A_5 \in A_2^{-}, \\
g(2) &= 3, & \text{because } & A_5, A_8, A_9 \in A_5^{-}, \\
g(3) &= 2, & \text{because } & A_8, A_9 \in A_6^{-}, \\
g(4) &= 3, & \text{because } & A_9, A_{12}, A_1 \in A_9^{-}, \\
g(5) &= 2, & \text{because } & A_{12}, A_1 \in A_{10}^{-}, \\
g(6) &= 3, & \text{because } & A_1, A_4, A_5 \in A_{13}^{-}.
\end{aligned}
$$

So the function $g(t)$ takes on values $\{2, 3, 2, 3, 2, 3\}$.

Every set of $A$'s and $B$'s, which gives rise to the same $g(t)$ and $h(t)$, is equivalent for the purposes of evaluating the inequality. In particular, we do not need to know what $m$ (the total number of subsystems $A_i$) is; only knowledge of $g(t)$ is enough. As an illustration, consider $m = 101$ and six members of a K-vector: $\{A_1, A_{10}, A_{31}, A_{59}, A_{60}, A_{82}\}$. They give rise to the same $g(t)$. Therefore, this selection of six $A$-members out of $m = 101$ will behave exactly the same as our first, $m = 13$ example.

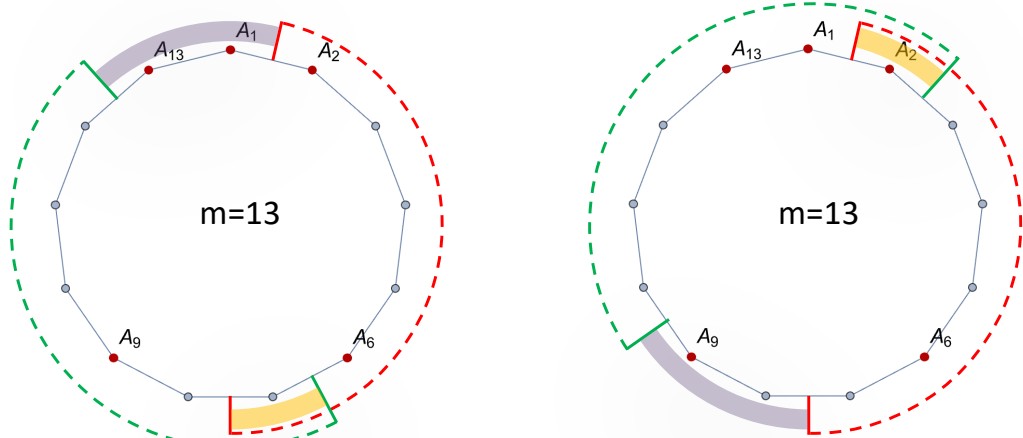

Figure 6: Illustrations of equation (A.12) (left) and equation (A.13) (right). In each case, the left hand side counts members in a region, which covers the whole $m$-gon except for a small overlap (yellow) and a small leftover (purple). On the left, the overlap is guaranteed to have no members while the leftover is guaranteed to have at least one member $A_{i_t}$. On the right, the leftover is guaranteed to have exactly one member $A_{i_{t+g(t)+1}}$ while the overlap may or may not contain members.

**Properties of $g(t)$**   They will be useful in the ensuing proofs. The definitions of $g(t)$ and $h(t)$ are identical, except that they describe $A$-type (respectively $B$-type) regions. Thus, although we phrase the discussion in terms of $g(t)$, the conclusions also apply to $h(t)$.

Once again, we stress that $g(t)$ implicitly depends on the $K(2p)$-vector. We call the number of $A$-members of the K-vector $k$; the examples above had $k = 6$ but different $m$. The domain of $g(t)$ is $\{1, 2 \ldots k\}$. We use $l = 2p - k$ for the number of $B$-members.

- The sum of $g(t)$ is:

$$\sum_{t=1}^{k} g(t) = \frac{k(k-1)}{2}. \tag{A.10}$$

  Therefore, the average of $g(t)$ is $(k-1)/2$. Equation (A.10) holds because for every pair of K-vector members $A_i$ and $A_j$, either $A_j \in A_{i+1}^-$ or $A_i \in A_{j+1}^-$. The right hand side counts the pairs.

- $g(t)$ cannot decrease in steps larger than 1. Say $A_{i+1}^-$ contains $g(t)$ members of the K-vector, the smallest-indexed among them being $A_j$. Then all but one of them are also in $A_{j+1}^-$, the only exception being $A_j$ itself.

$$g(t+1) \geq g(t) - 1. \tag{A.11}$$

  You can see this by comparing the red and green 'almost-semicircles' in Figure 5.

- The following inequalities hold because they describe two regions $A_{j+1}^-$, which 'almost' cover all $A_i$s once; see Figure 6. In the upper inequality, the $A_{j+1}^-$s overlap in a range that has no members of the K-vector, but leave out a range which contains at least one member ($A_{i_t}$ itself). In the lower inequality, the $A_{j+1}^-$s leave out a range that contains one member ($A_{i_{t+g(t)+1}}$) but overlap in a range, which might contain members.

$$g(t) + g(t + g(t)) \leq k - 1, \tag{A.12}$$

$$g(t) + g(t + g(t) + 1) \geq k - 1. \tag{A.13}$$

- Values taken on by $g(t)$:

$$0 \leq g(t) \leq k-1. \tag{A.14}$$

The extreme values 0 and $k-1$ are achieved only if all the members live in some common $A_j^+$, in which case the full range of $g(t)$ becomes $\{k-1, k-2, \ldots, 1, 0\}$.

- The possible behaviors for $g(t)$ for the smallest values of $k$ (up to cyclic reorderings) are:

$$
\begin{aligned}
k = 1: \quad & \{0\}, \\
k = 2: \quad & \{1, 0\}, \\
k = 3: \quad & \{2, 1, 0\} \text{ or } \{1, 1, 1\}, \\
k = 4: \quad & \{3, 2, 1, 0\} \text{ or } \{2, 2, 1, 1\}, \\
k = 5: \quad & \{4, 3, 2, 1, 0\} \text{ or } \{3, 3, 2, 1, 1\} \text{ or } \{3, 2, 2, 1, 2\} \text{ or } \{2, 2, 2, 2, 2\}.
\end{aligned}
\tag{A.15}
$$

- In terms of $g(t)$ and $h(t)$, the value of a toric inequality on a K($2p$)-vector with $k$ $A$-members and $2p - k$ $B$-members is:

$$\#\{(t, t') \text{ such that } g(t) + h(t') = p-1\} - \min\{k, 2p-k\}. \tag{A.16}$$

We emphasize that this expression is independent of $m$ and $n$.

### A.2 Proofs of lemmas in section 3

**Proof of Lemmas 1 and 2** The lemmas are about K4-vectors so $p = 2$. We use equation (A.16) and consult list (A.15) for different splits of $2p = k + l$:

- $(k, l) = (4, 0)$ or $(0, 4)$. — Vanishes directly from (A.1).

- $(k, l) = (3, 1)$ or equivalently $(1, 3)$. — Since $h(t) = 0$, we are effectively counting the number of 1's in $g(t)$, minus one (because $l = 1$). This gives 0 when $g(t)$ is $\{2, 1, 0\}$ and 2 when it is $\{1, 1, 1\}$.

- $(k, l) = (2, 2)$ — Here $g(t)$ and $h(t)$ are $\{0, 1\}$, which can add up to 1 in two ways: $0 + 1$ and $1 + 0$. Thus, equation (A.16) gives $2 - 2 = 0$.

**Proof of Lemma 3** Without loss of generality, we have $g(1) = g(2) = g(3) = 1$ and $h(1) = 0$ because only this structure (or one with $A \leftrightarrow B$ exchanged) gives a non-saturating K4-vector. We have two cases to consider:

$$
\begin{aligned}
&\text{Case (1):} \quad \text{adjoining an } A\text{-region with weight } 2, \\
&\text{Case (2):} \quad \text{adjoining a } B\text{-region with weight } 2.
\end{aligned}
$$

In Case (1), the only non-vanishing quantity $I(A_{i_t} : B_{j_1} | A_{i_t+1}^- B_{j_1+1}^-)$ is:

$$I(A_{i_{\text{affected}}} : B_{j_1} | A_{i_{\text{affected}}+1}^- B_{j_1+1}^-) = 2. \tag{A.17}$$

Since in this case $S_{A_1 A_2 \ldots A_m} = 1$, the graph saturates inequality (A.1) as claimed.

In Case (2), the only non-vanishing conditional mutual informations are:

$$I(A_{i_t} : B_{j_{\text{affected}}} | A_{i_t+1}^- B_{j_{\text{affected}}+1}^-) = 2. \tag{A.18}$$

There are three of them, one for each $A_{i_t}$. Since in this case $S_{A_1 A_2 \ldots A_m} = 3$, the graph saturates inequality (A.1) as claimed.

**Proof of Lemma 6** Consider the vectors, which are constructed in Lemma 3 by starting from non-saturating AAAB-vectors. We must show that these vectors span a linear space whose dimension equals: dim{non-saturating AAAB-vectors} − 1.

In the proof of Lemma 4, we showed that a single application of Lemma 3 to a non-saturating AAAB-vector $\vec{v}$ produces an entropy vector of the form

$$\tfrac{1}{2}\vec{w} - \tfrac{1}{2}\vec{v} + \text{saturating}, \tag{A.19}$$

where $\vec{w}$ is some other non-saturating AAAB-vector. Modding out the saturating part, which is immaterial for the present purposes, and multiplying by 2 to ease the notation, we see that Lemma 3 produces vectors of the form:

$$\begin{pmatrix} \\ +1 \\ \\ -1 \\ \end{pmatrix}. \tag{A.20}$$

This is a vector of coefficients of non-saturating AAAB-vectors. All unfilled coefficients are zeroes. Our claim will follow if we can show that repeated applications of Lemma 3 can produce vectors like this with **any** placement of the entries +1 and −1.

This is what we show in the following:

- Any vector of form (A.20)—understood in the basis of non-saturating AAAB-vectors—is in the holographic entropy cone because it can be generated by repeated applications of Lemma 3.

The argument is phrased in terms of non-saturating AAAB-vectors, but for $n \geq 3$ the argument concerning non-saturating ABBB-vectors is identical.

Lemma 3 operates by adding a fifth member to the initial non-saturating AAAB-vector $\vec{v}$. The membership of the other non-saturating vector $\vec{w}$ in (A.19) is a subset of the resulting quintuple. Therefore, in a single application of Lemma 3, $\vec{v}$ and $\vec{w}$ (as K4-vectors) must have three members in common.

For example, consider:

$$\vec{v} = \vec{v}(A_1^1, A_2^1, A_3^1, B_1^1), \qquad \text{and} \qquad \vec{w} = \vec{v}(A_1^1, A_2^1, A_4^1, B_1^1). \tag{A.21}$$

These vectors can appear together in (A.19)—after a single application of Lemma 3—because they have three members in common. The difference between $\vec{v}$ and $\vec{w}$ is in the replacement $A_3 \rightarrow A_4$. In this way, we can represent single applications of Lemma 3 as **hops** such as $\{A_1, A_2, A_3, B_1\} \rightarrow \{A_1, A_2, A_4, B_1\}$. One hop changes one member in a quadruple and it is associated with one application of Lemma 3 and one vector (A.20).

The only other restriction on a hop is that the quadruple after the hop must still be non-saturating. We call such hops 'valid.' In terms of functions $g(t)$ and $h(t')$, we see that a valid hop is one that preserves the structure of $g(t)$ being $\{1, 1, 1\}$ and of $h(t')$ being $\{0\}$. In particular, any sequence of hops, which goes from a non-saturating AAAB-vector to a non-saturating ABBB-vector—where $g(t)$ is $\{0\}$ and $h(t')$ is $\{1, 1, 1\}$—would have to pass through an AABB-vector whose $g(t)$ and $h(t')$ are both $\{1, 0\}$. That would be an invalid hop because all AABB-vectors are saturating. This is the reason why non-saturating AAAB-vectors and ABBB-vectors do not mix under applications of Lemma 3.

Now consider a sequence of two hops, for example:

$$\vec{v} = \vec{v}(A_1^1, A_2^1, A_3^1, B_1^1) \quad \rightarrow \quad \vec{w} = \vec{v}(A_1^1, A_2^1, A_4^1, B_1^1),$$
$$\vec{w} = \vec{v}(A_1^1, A_2^1, A_4^1, B_1^1) \quad \rightarrow \quad \vec{u} = \vec{v}(A_1^1, A_2^1, A_4^1, B_2^1). \tag{A.22}$$

This represents two applications of Lemma 3, each with one underlying five-armed star graph. Because the holographic entropy cone is a convex cone, those two entropy vectors can be added together without leaving the cone. For the vectors in (A.22), this means:

$$\left(\tfrac{1}{2}\vec{w} - \tfrac{1}{2}\vec{v} + \text{saturating}\right) + \left(\tfrac{1}{2}\vec{u} - \tfrac{1}{2}\vec{w} + \text{saturating}\right) \quad \rightarrow \quad \left(\tfrac{1}{2}\vec{u} - \tfrac{1}{2}\vec{v} + \text{saturating}\right). \tag{A.23}$$

This vector is also of the form (A.20) and, by construction, it is also in the holographic entropy cone. However, the members of the K4-vectors $\vec{v}$ and $\vec{u}$ do not generally have three members in common. This happens if the first hop and the second hop switch a different region, for example if $\vec{v}$ and $\vec{u}$ are:

$$\vec{v} = \vec{v}(A_1^1, A_2^1, A_3^1, B_1^1), \qquad \text{and} \qquad \vec{u} = \vec{v}(A_1^1, A_2^1, A_4^1, B_2^1). \tag{A.24}$$

Even though $\vec{v}$ and $\vec{u}$ do not have three regions in common, vector $\vec{u} - \vec{v}$ is in the holographic entropy cone because there exists a sequence of valid hops from $\vec{v}$ to $\vec{u}$ (in this example, via $\vec{w}$). In summary, the existence of a sequence of valid hops from $\vec{v}$ to $\vec{u}$ is a sufficient condition for $\vec{u} - \vec{v}$ to be in the holographic entropy cone.

In effect, we have rephrased the highlighted statement above as:

- Any two non-saturating AAAB-vectors are connected by a sequence of valid single-region hops.

Hops on the $B$-side, which replace $B_j \rightarrow B_{j'}$, can reset the $B$-content of a non-saturating AAAB-vector in one shot. The less trivial component of the assertion is that the $A$-members of a non-saturating AAAB-vector can also be adjusted by a sequence of single-region hops. This is what we prove next.

Because the remaining part of the proof concerns only triples of $A$-regions, from now on a hop will mean a replacement $A_i \rightarrow A_{i'}$ in a triple $\{A_{i_1}, A_{i_2}, A_{i_3}\}$. Every triple is characterized by a function $g(t)$, which must be of the form $\{1, 1, 1\}$ for the resulting AAAB-vector to be non-saturating. Accordingly, we will say that a hop is valid if it preserves the property of the triple $\{A_{i_1}, A_{i_2}, A_{i_3}\}$ that its $g(t)$ function is $\{1, 1, 1\}$.

We have reached the final restatement of our problem. It suffices to show that:

- Any two triples $\{A_{i_1}, A_{i_2}, A_{i_3}\}$ and $\{A_{i_1'}, A_{i_2'}, A_{i_3'}\}$ whose $g(t)$ functions are $\{1, 1, 1\}$ are connected through a sequence of valid single-region hops. That is, one can go from any such triple to another by adjusting one region at a time, without ever losing the property that the $g(t)$ function is $\{1, 1, 1\}$.

This suffices to prove Lemma 6 and that is what we demonstrate below.

We prove the claim by describing an explicit sequence of hops algorithmically; see Figure 7. We start from an initial triple $\{A_{i_1}, A_{i_2}, A_{i_3}\}$ with $1 \leq i_1 \leq i_2 \leq i_3 \leq m$. Define distances $d_{12} = i_2 - i_1$, $d_{23} = i_3 - i_2$, and $d_{31} = i_1 + m - i_3$.

1. If $d_{12}$, $d_{23}$, $d_{31}$ are $\{m/3, m/3, m/3\}$ or $\{(m-1)/3, (m-1)/3, (m+2)/3\}$ or $\{(m+1)/3, (m+1)/3, (m-2)/3\}$ (depending on $m$ (mod 3)), proceed to step 2. Otherwise, consider the largest of the three quantities $|d_{12} - d_{23}|$, $|d_{23} - d_{31}|$, $|d_{31} - d_{12}|$. In case of a two-way tie, take either one of the two larger quantities.

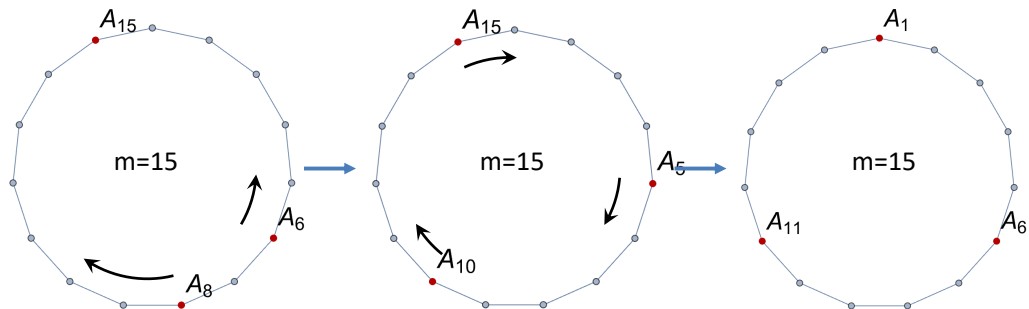

Figure 7: The algorithm by which any triple of $A$-regions can be deformed into configuration (A.25) without losing the property that its $g(t)$ function is $\{1,1,1\}$.

(a) Make a hop, which decreases the largest of the three quantities. For example, if $|d_{12} - d_{23}|$ is the largest and $d_{12} > d_{23}$ then the hop $A_{i_2} \to A_{i_2-1}$ will decrease that quantity by 2. On the other hand, if $|d_{12} - d_{23}|$ is the largest and $d_{12} < d_{23}$ then the hop $A_{i_2} \to A_{i_2+1}$ will decrease that quantity by 2.

(b) Repeat step 1(a) until $d_{12}$, $d_{23}$, $d_{31}$ become $\{m/3, m/3, m/3\}$ or $\{(m-1)/3, (m-1)/3, (m+2)/3\}$ or $\{(m+1)/3, (m+1)/3, (m-2)/3\}$, depending on $m \pmod 3$.

2. Repeat the series of hops $A_{i_1} \to A_{i_1+1}$, $A_{i_2} \to A_{i_2+1}$, $A_{i_3} \to A_{i_3+1}$ to achieve an arbitrary cyclic shift. If $m = 5$ or $7$, see special instructions in (A.29) below.

This algorithm connects any triple $\{A_{i_1}, A_{i_2}, A_{i_3}\}$ to one of the three canonical configurations:

$$\{A_1, A_{1+m/3}, A_{1+2m/3}\} \quad \text{or} \quad \{A_1, A_{1+(m-1)/3}, A_{1+2(m-1)/3}\}$$
$$\text{or} \quad \{A_1, A_{1+(m+1)/3}, A_{1+2(m+1)/3}\}. \tag{A.25}$$

If any initial triple can be brought to this canonical form then we can also reach any final triple, simply by running the sequence from the final triple to (A.25) backwards. What remains to be shown is:

• In the algorithm above, when step 1(a) or one hop in step 2 is applied to a triple $\{A_{i_1}, A_{i_2}, A_{i_3}\}$ whose $g(t)$ is $\{1,1,1\}$, the resulting triple after the hop also has $g(t)$ of the form $\{1,1,1\}$.

To verify this, observe the following:

$$g(t) \text{ is } \{1,1,1\}, \quad \text{if and only if} \quad \max\{d_{12}, d_{23}, d_{31}\} \leq (m-1)/2,$$
$$g(t) \text{ is } \{2,1,0\}, \quad \text{if and only if} \quad \max\{d_{12}, d_{23}, d_{31}\} \geq (m+1)/2. \tag{A.26}$$

These equivalences follow directly from the definition of $g(t)$. Now, step 1(a) in the algorithm never increases $\max\{d_{12}, d_{23}, d_{31}\}$. As for step 2, in the worst case it can increase:

$$d_{31} = (m+2)/3 \quad \to \quad (m+5)/3. \tag{A.27}$$

According to (A.26), this could potentially turn $g(t)$ into the form $\{2,1,0\}$ only if

$$(m+5)/3 \geq (m+1)/2, \tag{A.28}$$

that is if $m \leq 7$. In these special cases, we can effect the cyclic shift in step 2 like so:

$$m = 5: \quad \{A_1, A_3, A_5\} \to \{A_2, A_3, A_5\} \to \{A_2, A_4, A_5\} \to \{A_2, A_4, A_1\},$$
$$m = 7: \quad \{A_1, A_3, A_5\} \to \{A_1, A_3, A_6\} \to \{A_1, A_4, A_6\} \to \{A_2, A_4, A_6\}. \tag{A.29}$$

These hops never get to $\max\{d_{12}, d_{23}, d_{31}\} \geq (m+1)/2$. This completes the proof of Lemma 6.

**Proof of Lemma 8** The members of the non-saturating $K(2p)$-vector include $k$ regions $A_i$ and $l = 2p - k$ regions $B_j$. Without loss of generality, we assume $k \geq l$. We are working with K6- and higher K-vectors, so $p \geq 3$.

Using equation (A.1) but rephrasing in terms of $i_t$ ($i_t$ are the indices of the members of the K-vector, with $1 \leq t \leq k$), the value of the inequality on a K-vector is:

$$\sum_{t'=1}^{l} \left( \left( \sum_{t=1}^{k} \frac{1}{2} I(A_{i_t} : B_{j_{t'}} | A_{i_t+1}^- B_{j_{t'}+1}^-) \right) - 1 \right). \tag{A.30}$$

The lemma asks us to reset to $2p - 3$ the weight of one member of the K-vector, which we are free to choose. We will reset one of the $k$ $A$-members of the K-vector—that is, on the side that has the majority (or half) of its membership. This way, the resetting does not affect the term $-S_{A_1 A_2 ... A_m} = -S_{B_1 B_2 ... B_n} = -l$ and expression (A.30) is still valid.

**Idea of proof** The idea of the proof is that $2p - 3$ is a sufficiently large weight so that all the conditional mutual informations in (A.30) vanish, except those where $A_{i_t}$ is at the affected bond. Then, if we can also show

$$I(A_{i_{\text{affected}}} : B_{j_{t'}} | A_{i_{\text{affected}}+1}^- B_{j_{t'}+1}^-) = 2, \tag{A.31}$$

then we will have demonstrated that the resulting vector saturates the inequality. From here on we call the subsystem with the altered bond weight $A_{i_{\text{affected}}} \equiv A_{i_{t*}}$.

To inspect the conditional mutual information in (A.31), we write down all the entropies it involves using the functions $g(t)$ and $h(t')$ defined in (A.8-A.9):

$$S_{A_{i_{t*}+1}^- B_{j_{t'}+1}^-} = \min\{g(t^*) + h(t'), 4p - 4 - g(t^*) - h(t')\},$$
$$S_{A_{i_{t*}}^+ B_{j_{t'}+1}^-} = \min\{2p - 3 + g(t^*) + h(t'), 2p - 1 - g(t^*) - h(t')\},$$
$$S_{A_{i_{t*}+1}^- B_{j_{t'}}^+} = \min\{1 + g(t^*) + h(t'), 4p - 5 - g(t^*) - h(t')\}, \tag{A.32}$$
$$S_{A_{i_{t*}}^+ B_{j_{t'}}^+} = \min\{2p - 2 + g(t^*) + h(t'), 2p - 2 - g(t^*) - h(t')\} = 2p - 2 - g(t^*) - h(t').$$

The right hand sides are all of the form $\min\{$quantity$, (4p - 4) - $quantity$\}$ because after the adjustment of the special weight, the combined weight of all bonds in the star graph becomes $4p - 4$.

We now substitute expressions (A.32) into (A.31). This gives one equation with one unknown variable $g(t^*) + h(t')$, which involves four minima. A little algebra reveals that the equation is solved by the range:

$$(A.31) \text{ holds} \iff 1 \leq g(t^*) + h(t') \leq 2p - 3 = (k - 1) + (l - 1) - 1.$$

Using (A.14), we see that this is automatically satisfied in all circumstances except possibly in one special case: when $g(t)$ and $h(t')$ take on the extreme forms $\{k-1, k-2 \ldots 0\}$ and $\{l-1, l-2 \ldots 0\}$. However, in that scenario the underlying K-vector saturates the inequality, as is readily seen from (A.16). Since Lemma 8 concerns non-saturating K-vectors, this circumstance in fact never arises.

This establishes that equation (A.31) holds as an identity. In other words, it does not impose any conditions on which member of the initial K-vector can have its bond weight altered. As far as condition (A.31) is concerned, any choice will do.

It remains to verify the other condition mentioned in **Idea of proof** above. We state it as a separate lemma:

**Lemma 17** *Assume $p \geq 3$. In any non-saturating K(2p)-vector with $k \geq p$ A-members and $l = 2p - k \leq p$ B-members, there exists a choice of member $A_{i_{t*}}$ such that resetting its weight to $2p - 3$ causes*

$$I(A_{i_t} : B_{j_{t'}} | A^-_{i_t+1} B^-_{j_{t'}+1}) = 0 \,, \tag{A.33}$$

*for all $t \neq t^*$ (for all members $A_{i_t}$ whose bond weight was not altered).*

**Proof of Lemma 17** If we subdivide $A_{i_{t*}}$ into $2p - 3$ smaller subregions of weight 1 (maintaining the structure of a star graph) then quantity (A.33) becomes the conditional mutual information of one single region and another single region in a K-vector with membership equal to $4p - 4$. This is non-vanishing only if the conditioning region contains exactly $2p - 3$ members. Returning to the graph in the lemma, this means that (A.33) can fail only if the members of the graph are distributed like so:

$$\begin{aligned} \text{either}: \quad & A_{i_{t*}} \in A^-_{i_t+1} B^-_{j_{t'}+1} && \text{and all others in } \overline{A^+_{i_t} B^+_{j_{t'}}} \,, \\ \text{or}: \quad & A_{i_{t*}} \in \overline{A^+_{i_t} B^+_{j_{t'}}} && \text{and all others in } A^-_{i_t+1} B^-_{j_{t'}+1} \,. \end{aligned} \tag{A.34}$$

Rephrased in terms of functions $g(t)$ and $h(t')$, (A.34) requires $t$ and $t'$ such that:

$$\begin{aligned} \text{either}: \quad & t = t^* - 1 \quad \text{and} \quad g(t) = 1 \quad \text{and} \quad h(t') = 0 \,, \\ \text{or}: \quad & t = t^* + 1 \quad \text{and} \quad g(t) = k - 2 \quad \text{and} \quad h(t') = l - 1 \,. \end{aligned} \tag{A.35}$$

We must prove that these conditions—which imply $I(A_{i_t} : B_{j_{t'}} | A^-_{i_t+1} B^-_{j_{t'}+1}) \neq 0$—never arise if we choose $t^*$ wisely.

Conditions (A.35) require $h(t')$ to take on the extreme form $\{l-1, l-2 \ldots 0\}$, i.e. all $B$-members must be contained in some $B^+_j$. On the other hand, we have already seen that if $g(t)$ and $h(t')$ both take on the extreme forms $\{k-1, k-2 \ldots 0\}$ and $\{l-1, l-2 \ldots 0\}$ then the underlying K-vector saturates the inequality. This restricts the set of functions $g(t)$ under consideration to those with $\max g(t) = k - 2$. Note that $g(t) = 1$ implies $g(t+2) \geq k - 2$ by (A.13).

Our task is to prove that for any $g(t)$, which achieves $k - 2$ but not $k - 1$, we have:

$$\exists\, t^* \text{ such that } g(t^* - 1) \neq 1 \text{ and } g(t^* + 1) \neq k - 2 \,. \tag{A.36}$$

This is a very weak condition, which at large $k$ can be satisfied in multiple ways. We invite the reader to find a pattern among all functions $g(t)$ such that $\max g(t) = k - 2$, which makes the statement manifest.

To complete the proof formally, we give the following algebraic argument. Because $g(t)$ achieves both $k - 2$ and 1, and because it can never decrease in steps larger than 1 (inequality A.11), it must achieve every value between 1 and $k - 2$ (inclusive) at least once. To maintain (A.10), it must be that the full range of $g(t)$ contains exactly one additional pair $(q, k-1-q)$. In total, $g(t)$ can achieve values $\{1, k-2\}$ at most four times. Each time $g(t) = 1$ or $k - 2$, conditions (A.36) exclude either $t + 1$ or $t - 1$ from acting as $t^*$. That is, at most four values of $t$ are prevented from becoming $t^*$. Thus, if $k \geq 5$, a $t^*$ that satisfies (A.36) exists simply by counting.

It remains to inspect the cases $k = 4$ and $k = 3$. When $k = 4$, we have the unique (up to cyclic relabeling) pattern:

$$g(1) = 2 \,, \qquad g(2) = 2 \,, \qquad g(3) = 1 \,, \qquad g(4) = 1 \,. \tag{A.37}$$

We choose $t^* = 2$. When $k = 3$, we have the unique pattern $g(1) = g(2) = g(3) = 1$, which appears problematic. Recall, however, that $l \leq k$ and $k + l \geq 6$, so we also have $l = 3$.

Moreover, we have already concluded that conditions (A.35) can arise only if the $B$-regions are concentrated in one $B_j^+$, i.e. when $h(t')$ takes on values $\{2, 1, 0\}$. Thus, the K-vector with functions $g(t)$ and $h(t')$ given by $\{1, 1, 1\}$ and $\{2, 1, 0\}$ saturates the toric inequality, as is seen from equation (A.16). This completes the proof of Lemma 17 and Lemma 8.

**Proof of Lemma 10**   In the main text above Lemma 10 we established that non-saturating AAAB-vectors and ABBB-vectors can simultaneously appear in expansion (24) only if $k = l = 3$. The preceding paragraph—at the conclusion of the proof of Lemma 17—says that such K6-vectors with $k = l = 3$ are non-saturating only if the functions $g(t)$ and $h(t')$ are both of the form $\{1, 1, 1\}$. This is what we assume from now on.

Without loss of generality, we set the weight of the bond at $A_{i_1}$ to 3. Expansion (24) contains $\binom{6}{4} = 15$ K4-vectors. Among those, there is exactly one K4-vector of ABBB type, which contains $A_{i_1}$; its coefficient in expansion (24) is $+1/3$. On the other hand, there are exactly two K4-vectors of ABBB-type, which do not contain $A_{i_1}$ because they contain $A_{i_2}$ and $A_{i_3}$ instead; their coefficients are $-1/6$. All these vectors are non-saturating, as can be seen by applying (A.16) to $g(1) = 0$ and $h(1) = h(2) = h(3) = 1$. The claim is true because

$$1 \times \tfrac{1}{3} + 2 \times \left(-\tfrac{1}{6}\right) = 0. \tag{A.38}$$

Of course, if we alter the weight of a $B_j$ region then the coefficients of the non-saturating AAAB-vectors will add up to zero.

**Proof of Lemma 11**   The proof of Lemma 8 makes manifest that the claim is true for K6-vectors with $k = l = 3$, such that $g(t)$ and $h(t)$ are both $\{1, 1, 1\}$. This can be achieved for all values of $m$ and $n$. Specifically, given any $m$, choosing the three $A$-members of the K-vector to be $\{A_{(m-1)/2}, A_{(m+1)/2}, A_m\}$ produces $\{1, 1, 1\}$ for $g(t)$.

## A.3   Proofs of lemmas in section 4

We start with a rewriting of inequalities (11), which is analogous to (A.1):

$$\frac{1}{2} \sum_{i,j=1}^{m} I(A_i : B_{i+j} | A_{i+1}^{(j-1)} B_{i+j+1}^{(m-j)}) \geq S_{A_1 A_2 \dots A_m}. \tag{A.39}$$

We again consider a K($2p$)-vector with $k$ $A$-regions and $l$ $B$-regions (so $k + l = 2p$). Without loss of generality we assume $k \leq l$ so that $S_{A_1 A_2 \dots A_m} = l$. In analogy to equation (A.4), we can rewrite the value of the inequality on the K-vector as:

$$-l + \#\{\text{pairs } (A_i, B_{i+j}) \text{ of K}(2p)\text{-members s.t. } A_{i+1}^{(j-1)} B_{i+j+1}^{(m-j)} \text{ contains } p - 1 \text{ members}\}. \tag{A.40}$$

**Proof of Lemma 13**   We work with the projective plane inequalities in the form (A.39). We consider three possibilities:

- Four $A$-members and zero $B$-members or vice versa. Such vectors automatically saturate (A.39).

- Three $A$-members (call them $A_{i_{1,2,3}}$ with $1 \leq i_1 < i_2 < i_3 \leq m$), and one $B$-member $B_{j_*}$. The three potentially non-vanishing conditional mutual informations in (A.39) become $I(A_{i_t} : B_{j_*} | A_{i_t+1} A_{i_t+2} \dots A_{j_*-1})$, with $t = 1, 2, 3$. They do not vanish if and only if the

conditioning region $A_{i_t+1}^{(j_*-i_t-1)}$ contains precisely one of the two remaining $A$-members, in which case their value is 2. Stated algebraically, this means

$$I(A_{i_1} : B_{j_*}|A_{i_1+1}^{(j_*-i_1-1)}) = 2 \qquad \Longleftrightarrow \qquad i_2+1 \le j_* \le i_3\,,$$
$$I(A_{i_2} : B_{j_*}|A_{i_2+1}^{(j_*-i_2-1)}) = 2 \qquad \Longleftrightarrow \qquad i_3+1 \le j_* \le m,\ \text{or}\ 1 \le j_* \le i_1\,,$$
$$I(A_{i_3} : B_{j_*}|A_{i_3+1}^{(j_*-i_3-1)}) = 2 \qquad \Longleftrightarrow \qquad i_1+1 \le j_* \le i_2\,, \tag{A.41}$$

and all other conditional mutual informations in (A.39) vanish. Since $S_{A_1A_2...A_m} = 1$, no matter the value of $j_*$, each of these K4-vectors saturates the inequality.

The argument is nearly identical for three $B$-members and one $A$-member.

- Two $A$-members (call them $A_{i_{1,2}}$ with $1 \le i_1 < i_2 \le m$) and two $B$-members $B_{j_{1,2}}$, so $S_{A_1A_2...A_m} = 2$. We have four potentially non-vanishing conditional mutual informations, which are $I(A_{i_t} : B_{j_{t'}}|A_{i_t+1}^{(j_{t'}-i_t-1)}B_{j_{t'}+1}^{(m+i_t-j_{t'})})$ with $t, t' \in \{1, 2\}$. Each of these conditional mutual informations is non-vanishing (and equal to 2) if and only if the conditioning region

$$A_{i_t+1}^{(j_{t'}-i_t-1)}B_{j_{t'}+1}^{(m+i_t-j_{t'})} = A_{i_t+1}A_{i_t+2}\ldots A_{j_{t'}-1}B_{j_{t'}+1}B_{j_{t'}+2}\ldots B_{i_t}\,, \tag{A.42}$$

contains precisely one member of the K4-vector. In terms of the placement of $j_1$ and $j_2$, we have two cases to consider:

  (a) $i_1+1 \le j_1 < j_2 \le i_2$. Then $I(A_{i_1} : B_{j_1}|\ldots)$ and $I(A_{i_2} : B_{j_2}|\ldots)$ equal 2 and the two others vanish. Inequality (A.39) is saturated: $\frac{1}{2}(2+2)-2 = 0$.
  (b) $i_1+1 \le j_1 \le i_2$ but $j_2$ is outside that interval. Then all four conditional mutual informations $I(A_{i_{1,2}} : B_{j_{1,2}}|\ldots) = 2$ and the inequality evaluates to $\frac{1}{2}(2+2+2+2)-2 = 2$.

There is also the case where neither $j_1$ nor $j_2$ satisfies $i_1+1 \le j \le i_2$, but it is equivalent to case (a) by a reindexing of regions.

In summary, we have found that K4-vectors with structure AAAA, AAAB, ABBB, BBBB, and AABB (case (a) above) saturate inequality (11). On the other hand, K4-vectors with structure AABB (case (b) above) do not saturate it. On those vectors, the inequality evaluates to 2.

**Proof of Lemma 14** Consider a quadruple $\{A_{i_1}, A_{i_2}, B_{j_1}, B_{j_2}\}$ with $1 \le i_1 < i_2 \le m$ and assume

$$i_1+1 \le j_1 \le i_2\,, \qquad \text{and} \qquad i_2+1 \le j_2 \le i_1+m\,, \tag{A.43}$$

i.e. that the underlying K4-vector does not saturate the inequality. Now add a fifth member $B_{j_*}$ with weight 2. We stated below conditions (34) that choosing the fifth member to be a $B$-region does not implicate a loss of generality. Apply Lemma 4 to the resulting five-armed graph:

$$\vec{s}(B_{j_*}^2, A_{i_1}^1, A_{i_2}^1, B_{j_1}^1, B_{j_2}^1) = \ \tfrac{1}{2}\cancel{\vec{s}(B_{j_*}^1, A_{i_1}^1, B_{j_1}^1, B_{j_2}^1)} + \tfrac{1}{2}\cancel{\vec{s}(B_{j_*}^1, A_{i_2}^1, B_{j_1}^1, B_{j_2}^1)}$$
$$+ \tfrac{1}{2}\vec{s}(B_{j_*}^1, A_{i_1}^1, A_{i_2}^1, B_{j_1}^1) + \tfrac{1}{2}\vec{s}(B_{j_*}^1, A_{i_1}^1, A_{i_2}^1, B_{j_2}^1)$$
$$- \tfrac{1}{2}\vec{s}(A_{i_1}^1, A_{i_2}^1, B_{j_1}^1, B_{j_2}^1)\,. \tag{A.44}$$

The crossed terms are K4-vectors, which automatically saturate the inequality because their membership is ABBB. As for the vectors in the middle line, we claim that one of them saturates the inequality and the other one does not. This is true because (choosing the (mod $m$) range of the index $j_*$ appropriately) we have

$$\text{either}\ \ i_1+1 \le j_* \le i_2\,, \quad \text{or}\quad i_2+1 \le j_* \le i_1+m\,. \tag{A.45}$$

Therefore, either $(j_1, j_*)$ or $(j_2, j_*)$ fails conditions (33), but not both.

Finally, $\vec{s}(A_{i_1}^1, A_{i_2}^1, B_{j_1}^1, B_{j_2}^1)$ is the original K4-vector assumed in the lemma; it does not saturate the inequality. In the end, expansion (A.44) contains precisely two non-saturating K4-vectors, one with coefficient $+\frac{1}{2}$ and one with coefficient $-\frac{1}{2}$. Because the inequality evaluates to 2 on all non-saturating K4-vectors (we noted so in the proof of Lemma 13), it evaluates to zero on (A.44).

**Proof of Lemma 15** We begin by fixing one canonical non-saturating K4-vector with members $\{A_1, A_m, B_m, B_1\}$; call this entropy vector $\vec{v}_*$. We prove that for **any** non-saturating K4-vector $\vec{v} \neq \vec{v}_*$, the vector

$$\tfrac{1}{2}\vec{v} - \tfrac{1}{2}\vec{v}_* + \text{saturating}, \tag{A.46}$$

can be constructed by repeated applications of Lemma 14. Any such vector is linearly independent of all saturating vectors and of others constructed in the same way. Then the dimension of the linear space spanned by vectors (A.46) equals the count of all non-saturating K4-vectors $\vec{v}$ except for $\vec{v}_*$, which is what the lemma claims.

The mechanics is similar to the proof of Lemma 6. We will verify that any quadruple $\{A_{i_1}, A_{i_2}, B_{j_1}, B_{j_2}\}$ (with $1 \leq i_1 < i_2 \leq m$) that satisfies

$$i_1 + 1 \leq j_1 \leq i_2, \qquad \text{and} \qquad \left( i_2 + 1 \leq j_2 \leq m \quad \text{or} \quad 1 \leq j_2 \leq i_1 \right), \tag{A.47}$$

can be reached by a sequence of single-region hops originating from $\{A_1, A_m, B_m, B_1\}$, without ever losing property (A.47). This suffices because a hop corresponds to one application of Lemma 14. Adding up vectors, which are produced in each hop in the sequence $\vec{v}_* \to \ldots \to \vec{v}$, yields (A.46).

Let us represent quadruple $\{A_{i_1}, A_{i_2}, B_{j_1}, B_{j_2}\}$ in the form $(i_1, i_2; j_1, j_2)$. We wish to bring it to $(1, m; m, 1)$ by a sequence of changes, altering one index at a time, without ever losing property (A.47). An explicit sequence is:

$$(1, m; m, 1) \to (i_1, m; m, 1) \to (i_1, m; j_1, 1) \to (i_1, i_2; j_1, 1) \to (i_1, i_2; j_1, j_2). \tag{A.48}$$

Property (A.47) is preserved at every stage. This completes the proof of Lemma 15.

**Proof of Lemma 16** The proof is similar to that of Lemma 8. We consider a non-saturating K($2p$)-vector ($p \geq 3$) with $k$ members $A_{i_t}$ ($1 \leq t \leq k$) and $l = 2p - k$ members $B_{j_{t'}}$ ($1 \leq t' \leq l$). Without loss of generality, we assume $k \geq l$, which also implies $k \geq 3$. This assumption means $S_{A_1 A_2 \ldots A_m} = l$. We will modify the weight of the bond at a region $A_{i_{t_*}}$, which does not affect $S_{A_1 A_2 \ldots A_m} = l$.

We rewrite the value of the inequality in (A.39) in the form analogous to (A.30):

$$\sum_{t'=1}^{l} \left( \left( \sum_{t=1}^{k} \frac{1}{2} I(A_{i_t} : B_{j_{t'}} | A_{i_t+1}^{(j_{t'}-i_t-1)} B_{j_{t'}+1}^{(m+i_t-j_{t'})}) \right) - 1 \right), \tag{A.49}$$

see equation (A.42). As in the proof of Lemma 8, the idea is that resetting the bond weight at $A_{i_{t_*}}$ to $2p - 3$ causes all the conditional mutual informations in (A.49) to vanish except those where $t = t^*$, and the latter all equal 2. These conditions will set (A.49) to zero, thereby proving the claim. For easy reference, we spell out these sufficient conditions below:

$$\text{for all } t': \qquad I(A_{i_{t_*}} : B_{j_{t'}} | A_{i_{t_*}+1}^{(j_{t'}-i_{t_*}-1)} B_{j_{t'}+1}^{(m+i_{t_*}-j_{t'})}) = 2, \tag{A.50}$$

$$\text{for all } t \neq t_* \text{ and all } t': \qquad I(A_{i_t} : B_{j_{t'}} | A_{i_t+1}^{(j_{t'}-i_t-1)} B_{j_{t'}+1}^{(m+i_t-j_{t'})}) = 0. \tag{A.51}$$

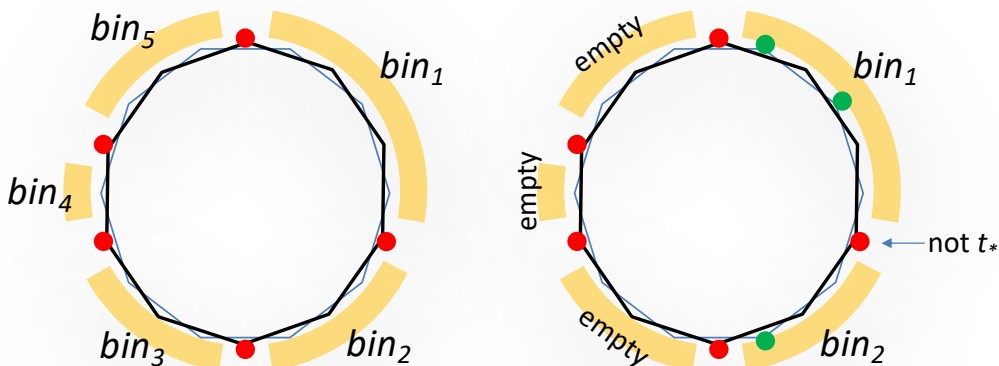

Figure 8: Left: The $A$-members divide locales for $B$-members into bins; see equation (A.52). Right: When precisely two bins are non-empty, one $A_{i_t}$ cannot be chosen as $A_{i_{t_*}}$ in Lemma 16. Red dots represent members $A_{i_t}$ while green dots represent members $B_{j_{t'}}$.

Consider the set of indices of the $A$-members $\{i_t\}_{t=1}^k$. The conditioning regions in (A.49) naturally partition the indices $j_{t'}$ of the $B$-members into bins $\mathcal{B}_t$:

$$\mathcal{B}_t = \{i_t + 1, i_t + 2, \ldots i_{t+1}\} \qquad \text{(understood (mod } m\text{))}. \qquad \text{(A.52)}$$

The division is illustrated in Figure 8. We prove the following statements:

1. If all $B$-members have indices in one bin $\mathcal{B}_t$ and all other bins are empty then the underlying K-vector saturates the inequality. Therefore, we need only consider cases where the indices of the $B$-members fall in at least two bins.

2. Suppose only two bins $\mathcal{B}_t$ and $\mathcal{B}_{t+1}$ are populated. (That is, only two bins are non-empty and they are consecutive.) Choose any $t_* \neq t+1$ (see Figure 8 for illustration) and reset the bond weight at $A_{i_{t_*}}$ to $2p-3$. Then equations (A.50-A.51) hold.

3. Otherwise, choose $t_*$ arbitrarily and reset the bond weight at $A_{i_{t_*}}$ to $2p-3$. Then equations (A.50-A.51) hold.

For statement 1 above, without loss of generality, let the indices of the $A$-members be

$$1 = i_1 < i_2 < \ldots < i_k \leq m, \qquad \text{(A.53)}$$

and the indices of the $B$-members be

$$2 \leq j_1 < j_2 < \ldots j_l \leq i_2. \qquad \text{(A.54)}$$

The conditional mutual information $I(A_{i_t} : B_{j_{t'}} | A_{i_t+1}^{(j_{t'}-i_t-1)} B_{j_{t'}+1}^{(m+i_t-j_{t'})})$ is non-vanishing if and only if $A_{i_t+1}^{(j_{t'}-i_t-1)} B_{j_{t'}+1}^{(m+i_t-j_{t'})}$ contains precisely $p-1$ members, in which case it equals 2. Now region $A_{i_t+1}^{(j_{t'}-i_t-1)}$ contains $k-t+1$ $A$-members (for $t \neq 1$) and 0 members (for $t = 1$); we can capture this simply as $k-t+1$ by taking $2 \leq t \leq k+1$. Meanwhile, region $B_{j_{t'}+1}^{(m+i_t-j_{t'})}$ contains $l-t'$ $B$-members. Thus, using $k+l = 2p$, for a non-vanishing conditional mutual information we must have $t + t' = p + 2$. In effect, the value of the inequality on this K-vector is:

$$-l + \# \left\{ (t, t') \text{ such that } 2 \leq t \leq k+1 \text{ and } 1 \leq t' \leq l \text{ and } t = \tfrac{k+3}{2} - \left(t' - \tfrac{l+1}{2}\right) \right\}. \qquad \text{(A.55)}$$

Because $k \geq l$ by assumption, $t$ is in the desired range for every $1 \leq t' \leq l$ so the value of the inequality is zero, as claimed.

From now on we assume that the indices of the $B$-members fall in at least two bins. This directly implies condition (A.50). To see this, fix $t'$ and let $x$ be the number of members in $A_{i_{t_*}+1}^{(j_{t'}-i_{t_*}-1)}B_{j_{t'}+1}^{(m+i_{t_*}-j_{t'})}$. Observing that $0 \leq x \leq 2p-2$, compute:

$$\begin{aligned}
I(A_{i_{t_*}}&:B_{j_{t'}}|A_{i_{t_*}+1}^{(j_{t'}-i_{t_*}-1)}B_{j_{t'}+1}^{(m+i_{t_*}-j_{t'})})\\
&= \min\{x+1, 4p-5-x\} + \min\{x+2p-3, 2p-1-x\} - \min\{x, 4p-4-x\} - (2p-2-x)\\
&= (2p-2) - |x-1| - |(2p-3)-x|.
\end{aligned} \tag{A.56}$$

For any $1 \leq x \leq 2p-3$ this equals 2, as desired. The only values of $x$ that do not lead to (A.50) are $x=0$ or $x=2p-2$. In other words, the failure of (A.50) implies one of the following:

- $x=2p-2$. That is, $A_{i_{t_*}+1}^{(j_{t'}-i_{t_*}-1)}B_{j_{t'}+1}^{(m+i_{t_*}-j_{t'})}$ contains all the members other than $A_{i_{t_*}}$ and $B_{j_{t'}}$ while $A_{j_{t'}}^{(m-j_{t'}+i_{t_*})}B_{i_{t_*}+1}^{(j_{t'}-i_{t_*}-1)}$ contains no members.

- $x=0$. The other way around: $A_{j_{t'}}^{(m-j_{t'}+i_{t_*})}B_{i_{t_*}+1}^{(j_{t'}-i_{t_*}-1)}$ contains all the members other than $A_{i_{t_*}}$ and $B_{j_{t'}}$ while $A_{i_{t_*}+1}^{(j_{t'}-i_{t_*}-1)}B_{j_{t'}+1}^{(m+i_{t_*}-j_{t'})}$ contains no members.

In the $x=2p-2$ case, all $A$-members have indices $i_{t_*} \leq i_t \leq j_{t'}-1$ and all $B$-members have indices $j_{t'} \leq j_{s'} \leq i_{t_*}+m$. This means that all $B$-members live in one bin $\mathcal{B}_{t_*-1}$. In the $x=0$ case, all $A$-members have indices $j_{t'} \leq i_t \leq i_{t_*}$ and all $B$-members have indices $i_{t_*}+1 \leq j_{s'} \leq j_{t'}$. This means that all $B$-members live in one bin $\mathcal{B}_{t_*}$.

In summary, we have verified that in a non-saturating K-vector (whose $B$-members are necessarily split into at least two bins), changing the weight of any bond to $2p-3$ automatically implies equation (A.50). What remains is to prove the part of statements 2 and 3, which concerns condition (A.51). It is easier to prove the contrapositive. We state it as a separate lemma, which is analogous to Lemma 17:

**Lemma 18** *Take a non-saturating K-vector with $k$ members $A_{i_t}$ and $l$ members $B_{i_{t'}}$ ($k \geq l$ and $k+l \leq 6$) and change the weight of the bond at some $A_{i_{t_*}}$ to $2p-3$. If equation (A.51) does not hold for any $t, t'$ then all B-members live in two consecutive bins $\mathcal{B}_t$ and $\mathcal{B}_{t+1}$ and $t_* = t+1$.*

In other words, if after changing the weight of one $A$-member to $2p-3$ we still have a non-saturating vector, the special member must have been chosen in violation of Figure 8. Any other choice would have worked.

Observe that $I(A_{i_t}:B_{j_{t'}}|A_{i_t+1}^{(j_{t'}-i_t-1)}B_{j_{t'}+1}^{(m+i_t-j_{t'})}) = I(A_{i_t}:B_{j_{t'}}|A_{j_{t'}}^{(m-j_{t'}+i_t)}B_{i_t+1}^{(j_{t'}-i_t-1)})$. In other words, because the union of all $A_i$'s and $B_j$'s is in a pure state, two different conditioning regions—$A_{i_t+1}^{(j_{t'}-i_t-1)}B_{j_{t'}+1}^{(m+i_t-j_{t'})}$ or $A_{j_{t'}}^{(m-j_{t'}+i_t)}B_{i_t+1}^{(j_{t'}-i_t-1)}$—define the same conditional mutual information. By temporarily splitting $A_{i_{t_*}}$ into $2p-3$ independent subsystems of weight 1 (like we did in the proof of Lemma 17), we see that the said conditional mutual information is non-vanishing only if one of the two conditioning regions contains $A_{i_{t_*}}$ alone and all the other members live elsewhere. That is, we have two cases to consider:

$$\text{only } A_{i_{t_*}} \in A_{i_t+1}^{(j_{t'}-i_t-1)}B_{j_{t'}+1}^{(m+i_t-j_{t'})}, \text{ and all others fall in } \overline{A_{i_t+1}^{(j_{t'}-i_t-1)}B_{j_{t'}+1}^{(m+i_t-j_{t'})}}, \tag{A.57}$$

$$\text{only } A_{i_{t_*}} \in A_{j_{t'}}^{(m-j_{t'}+i_t)}B_{i_t+1}^{(j_{t'}-i_t-1)}, \text{ and all others fall in } \overline{A_{j_{t'}}^{(m-j_{t'}+i_t)}B_{i_t+1}^{(j_{t'}-i_t-1)}}. \tag{A.58}$$

Assuming (A.57), we see that all the $B$-members live in $\overline{B_{j_{t'}+1}^{(m+i_t-j_{t'})}} = B_{i_t+1}^{(j_{t'}-i_t)}$ and all $A$-members other than $A_{i_{t_*}}$ live in $\overline{A_{i_t+1}^{(j_{t'}-i_t-1)}} = A_{j_{t'}}^{(m-j_{t'}+i_t+1)}$. In other words, the indices $i_s$ of all $A$-regions other than $A_{i_{t_*}}$ satisfy

$$j_{t'} \leq i_s \leq i_t + m, \quad \text{for all } s \neq t_*, \tag{A.59}$$

while the indices $j_{s'}$ of all the $B$-members satisfy

$$i_t + 1 \leq j_{s'} \leq j_{t'}, \quad \text{for all } s'. \tag{A.60}$$

If there were no $A_{i_{t_*}}$, property (A.60) would mean that all $B$-members live in the bin $\mathcal{B}_t$. With $A_{i_{t_*}}$ included, they are divided between $\mathcal{B}_t$ and $\mathcal{B}_{t_*}$ while all other bins are empty. Moreover, the bins $\mathcal{B}_t$ and $\mathcal{B}_{t_*}$ are necessarily consecutive, so $t_* = t + 1$, which is what we wanted to prove.

Finally, assume (A.58). Then all the $B$-members live in $\overline{B_{i_t+1}^{(j_{t'}-i_t-1)}} = B_{j_{t'}}^{(m-j_{t'}+i_t+1)}$ and all $A$-members other than $A_{i_{t_*}}$ live in $\overline{A_{j_{t'}}^{(m-j_{t'}+i_t)}} = A_{i_t}^{(j_{t'}-i_t)}$. In other words, the indices $i_s$ of all $A$-regions other than $A_{i_{t_*}}$ satisfy

$$i_t \leq i_s \leq j_{t'} - 1, \quad \text{for all } s \neq t_*, \tag{A.61}$$

while the indices $j_{s'}$ of all the $B$-members satisfy

$$j_{t'} \leq j_{s'} \leq i_t + m, \quad \text{for all } s'. \tag{A.62}$$

Combining the two conditions we find that all $B$-members have indices between $i_s + 1 \leq j_{s'} \leq i_t + m$ (for all $s \neq t_*$). If there were no $A_{i_{t_*}}$, this would mean that all $B$-members live in one bin, which is indexed by the largest $s$ in (A.61). The inclusion of $A_{i_{t_*}}$ reveals that that non-empty bin is indexed by $s = t - 2 \pmod{l}$. It also splits up the non-empty bin into two bins $\mathcal{B}_{t-2}$ and $\mathcal{B}_{t_*} = \mathcal{B}_{t-1}$.

This establishes Lemma 18 and therefore Lemma 16.

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
