# Peer review of "Two infinite families of facets of the holographic entropy cone"

_SciPost Physics, doi:SciPost Phys. 17, 084 (2024)_

## Round 1 · Referee Report · Anonymous (Referee 1) · 2024-5-6

Strengths

1) This paper proves an interesting result in full technical detail. 2) The paper makes good decisions about which portions of the proof to relegate to the appendices, and which lemmas are important to the understanding of the proof and thus need to be in the main text. 3) The paper nicely puts their result in the context of the larger program to better understand and describe the holographic entropy cone. 4) The proof is mathematically solid, and its method provides a path for a possible interpretation of the toric family of inequalities in terms of 4- and 6- party entaglement. 5) The discussion nicely relates to the physics content of the inequalities at issue; the uniformly-weighted star graphs corresponded to old black holes with each leg corresponding to a previously emitted Hawking quantum. This reader will be interested to see further understanding along these lines.

Weaknesses

All weaknesses are minor and easily remediable as per the requested changes below.

Report

This paper proves that two infinite families of holographic entropy inequalities are as tight as possible. That is, for each inequality, they construct $2^N-2$ linearly independent saturating holographic entropy vectors, showing that the achievable saturating surface is codimension one in the $2^N-1$ dimensional entropy space.

The result is of interest first because it completes the mathematical understanding of this set of inequalities (by understanding when they can be saturated). It improves the understanding of the holographic cone, as some previous facets now have a generalization as facets. And most importantly, its relation to understanding the nature of old black holes, as well as to entanglement in four and six party arrangements, is likely to produce interesting results in the future.

As the abstract admits, the proof itself is technical, but the paper does a good job of presenting its main ideas in the body while relegating more technical details to the appendix. The techniques used in the proof are also relevant; as pointed out in the conclusion, the fact that only star-graphs are needed, and the interplay between the $K4$ and $K6$ vectors, may point to an interpretation of these inequalities in terms of 4 and 6 party entanglement.

Requested changes

1) Apparent typo in right hand side of equation 3.2, one argument reads $X_0^1, X_1^1, X_2^2,X_4^1$ but should read $X_0^1,X_1^1,X_2^1,X_4^1$. 2) Figure 5 in the appendix should say that $g(1)$ is marked in red while $g(2)$ is marked in green. 3) Occasional unusual phrasings , e.g. "How to decide on that?" or "No because otherwise".

Recommendation

Publish (easily meets expectations and criteria for this Journal; among top 50%)

---

## Round 1 · Referee Report · Anonymous (Referee 2) · 2024-8-11

Report

The paper demonstrates that two infinite families of holographic entropy inequalities constructed in ref.[13] are in fact facets of the holographic entropy cone (as opposed to redundant inequalities obtainable as a positive combination of other holographic inequalities). The proof method consists of explicit construction of a sufficiently large collection of star graphs (mostly perfect tensors, supplemented by "flower graphs") whose entropy vectors are linearly independent and saturate the inequalities. Although a vested reader might glean further insight into the inequalities from these derivations, the paper does not offer particularly illuminating lessons beyond a heuristic observation regarding distribution of entanglement. As such, I think the main value for the broader readership is the statement itself, i.e. the verification of facetness. This contributes a new result to the holographic entropy cone story. The paper meets the acceptance criteria and thus merits publication.

Requested changes

Here are optional suggestions regarding the presentation, in order of decreasing importance.

1- The introduction of facets could be made clearer.

(a) First, I would suggest specifying the characterization of inequalities as taking the form $\text{LHS}_i \ge \text{RHS}_i$ before eq.(1.1), and preferably before marginal consistency is specified. In other words, I would find it more logical to have (1.3) and (1.4) precede (1.1).

(b) Second, marginal consistency (1.1) should be explicitly supplemented with all other holographic entropy inequalities being satisfied (otherwise it would fail consistency in the first place); perhaps the authors also wish to further distinguish $\text{LHS}_j \ge \text{RHS}_j$ from $\text{LHS}_j > \text{RHS}_j$ for all $j\ne i$. Indeed, the phrase "hyperplanes of marginal consistency" below (1.4) would (by conventional usage; see point 5) imply the latter, in which case I'd suggest further refinement of (1.1) replacing "for some $i$" with "for a single $i$."

(c) Finally, I think it's important to already indicate explicitly the key point that a requirement for a facet is not merely a true inequality, but also a non-redundant one. In other words, the characterization of (1.5) does not suffice without this additional characterization of the collection $\text{LHS}_i \ge \text{RHS}_i$. Otherwise, it may either not be possible to saturate the true inequality anywhere inside the cone (except for the origin), or it may only be possible to saturate it on a higher ($>1$) codimension locus. In particular, it might be pedagogically opportune after (1.5) to mention, in contrast to MMI, other inequalities (such as SSA) which are not facets.

2- The first bullet point on p.3, "They subsume many known facets of the holographic entropy cone, which were discovered in [8–11]." is rather vague. It would seem more natural to indicate how many, or the fraction of a specified number, and perhaps even which ones (or refer to the specification on p.5). Also, the reader may wonder whether or not the inequalities found in [12] are included in this statement, or why this work is not referenced among the list known inequalities here.

3- The statement on p.3 that "As all previously known facets are isolated (not part of any known infinite families)..." could be accompanied by a clarifying parenthetical remark regarding the dihedral family (e.g., up to how many parties was facetness established previously), to forestall the reader's potential confusion of why these are not a counter-example to the assertion.

4- The Lemmas are not always phrased in a form of a mathematical statement; some are accompanied by proofs, some by examples/comments, some by definitions. Adhering to a more conventional usage of Lemma might seem cleaner.

5- The usage of the word "hyperplane" is unconventional. Typically, a hyperplane is defined as a codimension-1 subspace, whereas the authors co-opt this for higher codimensions as well.

6- Just above Lemma 9 (p.10) in the sentence "Our next task ... codimension-2 hyperplane" it could be worth to re-specify "in the span of K4-vectors."

7- There are two occurrences of $\mathbb{RP}^2$ (on pp.17 and 18) which look unnecessarily technical -- one could either define this already as a name for (2.4), or just stick to "projective plane" throughout.

Recommendation

Publish (meets expectations and criteria for this Journal)

---

## Round 4 · Author Response

We thank the reviewers for their very careful reading of the manuscript and for the positive feedback.
We have implemented all changes suggested by the reviewers.
We have implemented all changes suggested by the reviewers.

---

## Round 4 · List of Changes

Changes suggested by Referee 1 (report from 2024-5-6):
(1) A typo - We fixed it. Thank you for noticing it!
(2) We have significantly expanded the caption of Figure 5, including the explanation suggested by the Referee.
(3) Unusual phrasings - We changed four arguably extravagant phrasings, including the two listed by the Referee. They can be found in the source text marked with %###A3
Changes suggested by Referee 2 (report from 2024-the 8-11):
(1) We have completely rewritten the first 1.5 pages of the Introduction, following the route suggested by the Referee.
(a) We now introduce the holographic entropy cone before marginality, as suggested by the Referee.
(b) We now specify that marginality wrt one inequality requires consistency with all other inequalities, see the new equation (1.6). The referee further suggested writing "for a single i" instead of "for some i" in that equation. We retained "for some i" but added an explanatory sentence in the text leading up to (1.6). The new sentence reads: "Less generic ways to achieve marginality occur on intersections of facets, which form higher codimension loci in entropy space." We think this sentence should avert any confusion.
(c) We emphasize non-redundancy and discuss strong subadditivity as an example of a redundant inequality, exactly as suggested by the Referee.
(2) We now give a comprehensive account of which previously known facets are subsumed by our infinite families. We have also added a footnote, which explains why there is no overlap between the facets we verify (N odd) and those discovered in [14] (N=6; Ref. [12] in the previous version). A minor explanation: the mismatch between the assumed values of N is the reason why we previously had not listed Ref. [14] in this passage.
(3) We have added a footnote, which explains what was previously known about the facetness of the dihedral inequalities.
(4) Lemmas not in a mathematical form - We believe we fixed this. What was originally written as Lemma 1, 2, 3 had the least lemma-like appearance; we now cast the same assertions as Fact 1, 2, 3. In the new Lemma 2 (previously Lemma 5), we separated the definitions of "non-saturating AAAB-vectors" and "non-saturating ABBB-vectors" from the actual lemma. In multiple lemmas, we changed the writing so that a clear "Then..." statement announces the predicate.
(5) We removed all instances of "codimension-2 hyperplane"; they are now "codimension-2 subspaces".
(6) We added "in the span of K4-vectors" exactly as suggested.
(7) We removed the two appearances of $\mathbb{RP}^2$ and used "projective plane inequalities" instead.
(1) A typo - We fixed it. Thank you for noticing it!
(2) We have significantly expanded the caption of Figure 5, including the explanation suggested by the Referee.
(3) Unusual phrasings - We changed four arguably extravagant phrasings, including the two listed by the Referee. They can be found in the source text marked with %###A3
Changes suggested by Referee 2 (report from 2024-the 8-11):
(1) We have completely rewritten the first 1.5 pages of the Introduction, following the route suggested by the Referee.
(a) We now introduce the holographic entropy cone before marginality, as suggested by the Referee.
(b) We now specify that marginality wrt one inequality requires consistency with all other inequalities, see the new equation (1.6). The referee further suggested writing "for a single i" instead of "for some i" in that equation. We retained "for some i" but added an explanatory sentence in the text leading up to (1.6). The new sentence reads: "Less generic ways to achieve marginality occur on intersections of facets, which form higher codimension loci in entropy space." We think this sentence should avert any confusion.
(c) We emphasize non-redundancy and discuss strong subadditivity as an example of a redundant inequality, exactly as suggested by the Referee.
(2) We now give a comprehensive account of which previously known facets are subsumed by our infinite families. We have also added a footnote, which explains why there is no overlap between the facets we verify (N odd) and those discovered in [14] (N=6; Ref. [12] in the previous version). A minor explanation: the mismatch between the assumed values of N is the reason why we previously had not listed Ref. [14] in this passage.
(3) We have added a footnote, which explains what was previously known about the facetness of the dihedral inequalities.
(4) Lemmas not in a mathematical form - We believe we fixed this. What was originally written as Lemma 1, 2, 3 had the least lemma-like appearance; we now cast the same assertions as Fact 1, 2, 3. In the new Lemma 2 (previously Lemma 5), we separated the definitions of "non-saturating AAAB-vectors" and "non-saturating ABBB-vectors" from the actual lemma. In multiple lemmas, we changed the writing so that a clear "Then..." statement announces the predicate.
(5) We removed all instances of "codimension-2 hyperplane"; they are now "codimension-2 subspaces".
(6) We added "in the span of K4-vectors" exactly as suggested.
(7) We removed the two appearances of $\mathbb{RP}^2$ and used "projective plane inequalities" instead.

---

## Editorial Decision

published